



# Modelling the atmospheric $^{34}$S-sulfur budget in a column model under volcanically quiescent conditions

Juhi Nagori[1], Narcisa Nechita-Bândă[1], Sebastian Oscar Danielache[2,3], Masumi Shinkai[2], Thomas Röckmann[1], and Maarten Krol[1,4]

[1]Institute of Marine and Atmospheric Research, University of Utrecht, Utrecht, The Netherlands
[2]Department of Material and Life Sciences, Faculty of Science & Technology, Sophia University
[3]Earth and Life Sciences Institute, Tokyo Institute of Technology
[4]Meteorology and Air Quality, Wageningen University, Wageningen, The Netherlands

*Correspondence to:* Juhi Nagori (j.v.nagori@uu.nl) and Maarten Krol (m.c.krol@uu.nl)

**Abstract.**

We investigated the sulfur isotope budget of atmospheric carbonyl sulfide (COS) and the role of COS as a precursor for stratospheric sulfate aerosols (SSA). Currently, the sulfur isotopic budgets for both SSA and tropospheric COS are unresolved. Moreover, there is some debate on the significance of COS on SSA formation. With the use of an atmospheric column model, we model the isotopic composition of COS to resolve some of the uncertainties in its budget. We attempt to constrain the isotopic budget ($^{32}$S and $^{34}$S) of COS in the troposphere and the stratosphere. We are able to constrain the model results to match the observed COS isotopic signature at the surface, which has recently been measured to lie between $\delta^{34}$S = 10–14 permil (‰). When we propagate this composition to SSA, we match the isotopic signal of SSA that was measured in volcanically quiescent times at 18 km as $\delta^{34}$S = 2.6 ‰. Our results show that COS becomes isotopically enriched during destruction in the stratosphere, and this enriched isotopic signal of COS propagates through $SO_2$ to sulfate, creating strong positive isotopic gradients of both $SO_2$ and sulfate in the lower stratosphere. Sensitivity tests indicate that the enriched sulfur in the stratosphere is mostly sensitive to COS photolysis, and to a lesser extent to biosphere uptake and COS emission signature. A better quantification of these processes could further support the role of COS in sustaining the SSA layer. Hence, there is a need for isotopic measurements for both stratospheric COS and SSA to better constrain these contributions.

## 1 Introduction

While $CO_2$ and other greenhouse gases in the troposphere warm the Earth, aerosols offset some of the global warming. There is a persistent layer of sulfur stratospheric aerosols (SSA) that is found between 13–30 km above the surface, called the Junge layer (Junge, 1966). By reflecting solar radiation, the SSA layer reduces the radiation that reaches the Earth's surface, thereby cooling the Earth (Junge and Manson, 1961; Crutzen, 1976). Sulfur dioxide ($SO_2$) from explosive volcanic eruptions perturbs this Junge layer, and has, historically, repeatedly caused global cooling by adding large amounts of sulfur to the stratosphere, for instance, after the Mount Pinatubo eruption in 1991 (McCormick et al., 1995; Rosenfield et al., 1997; Gao et al., 2008; Arfeuille et al., 2014).



In volcanically quiescent periods, the most abundant sulfur compound is carbonyl sulfide (COS or OCS) (Crutzen, 1976). COS is relatively inert and long-lived (2–4 years) in the troposphere, hence it is transported to the stratosphere (Brühl et al., 2012), where COS is converted to SSA. COS is oxidised in the atmosphere through chemical reactions with the hydroxyl

radical (OH), with oxygen radicals ($O^3P$), and through photolysis, a pathway that is only prominent in the stratosphere above the ozone layer (Hattori et al., 2012; Danielache et al., 2008; McKee and Wine, 2001). Another important pathway of COS removal is uptake by the biosphere (Protoschill-Krebs et al., 1996; Kettle, 2002; Berry et al., 2013). The COS budget, however, is still poorly constrained, with substantial uncertainties in sources and sinks (Whelan et al., 2018).

Other relevant sulfur gases in the atmosphere are $SO_2$, hydrogen sulfide ($H_2S$), dimethyl sulfide (DMS or $CH_3SCH_3$) and carbon disulfide ($CS_2$). In comparison to COS, these sulfur gases have a short lifetime and react quickly in the troposphere (Brühl et al., 2012). Of these gases, $CS_2$ is converted to the longer-lived COS with a yield of about 83% (Stickel et al., 1993). In contrast, $H_2S$ is mainly converted to $SO_2$, as is $CH_3SCH_3$. However, some $CH_3SCH_3$, in extremely pristine conditions, could be converted to COS, with a yield of 0.7% (Barnes et al., 1994, 1996; Albu et al., 2006).

In non-volcanic periods COS is often considered a major source of SSA (Notholt et al., 2003). However, $SO_2$ could be directly injected into the stratosphere through vigorous convection in the tropics, and could contribute to (non-volcanic) SSA formation as well (Kjellström, 1997). Hence the contribution of COS for SSA formation is debated. Chin and Davis (1995) calculated that only about 9% of COS that enters the stratosphere is converted to SSA, while Kjellström (1997) states that this

conversion is only 4%. More recent papers, however, consider the COS to SSA conversion to be higher. Pitari et al. (2002) estimated a stratospheric COS conversion fraction of about 43%. Modelling studies postulated stratospheric COS to SSA conversion as high as 56% − 70% (Brühl et al., 2012; Sheng et al., 2015). The rest of the SSA comes is considered to be from $SO_2$, though some uncertainties exist (Brühl et al., 2012).

Isotope measurements help resolve some of the uncertainties in the COS budget. Variations in isotopic composition can arise due to differences in sources and sinks. Hence, isotopic analysis is used to identify sources, and the effects of sinks and chemical transformations of atmospheric trace gases (Krouse and Grinenko, 1991; Brenninkmeijer, 2009). Most work has focused on the COS isotopic ratios of $^{34}S$ relative to $^{32}S$ from different sources and the fractionation associated with different conversion reactions. This ratio is commonly expressed relative to a standard ratio as a $\delta^{34}S$ value, expressed in ‰ (see Eq. 6). In an earlier

study, Krouse and Grinenko (1991) calculated a $\delta^{34}S$ value of COS in the troposphere of about +11 ‰. Recent measurements carried out in Japan for atmospheric $\delta^{34}S$ ranged from 9.7 to 14.5 ‰ (Kamezaki et al., 2019; Hattori et al., 2020). Measurements carried out in Israel and the Canary Islands led to background values of $\delta^{34}S$ around $13 - -14$‰ (Angert et al., 2019; Davidson et al., 2021). It has been assumed that this difference in measurements might be due to COS from anthropogenic origin (Kamezaki et al., 2019; Hattori et al., 2020). Indeed, more recent studies support that anthropogenic COS is associated with

lower $\delta^{34}S$ values compared to oceanic COS. A seasonal variation was observed in Japan which could be attributed to different





air masses arriving at the measurement location (Hattori et al., 2020). Air masses from the continent were more prominent in winter and the atmospheric COS measured was depleted i.e. with a smaller $\delta^{34}$ S value (Hattori et al., 2020). According to Davidson et al. (2021) the anthropogenic signature of COS was estimated at about 8 ‰, and the oceanic COS at about 15 ‰. The oceanic measurements were a composite of direct COS emissions (13 ‰), and oxidation from $CS_2$ (16 ‰) and $CH_3SCH_3$

(20 ‰) (Davidson et al., 2021). Lastly, Baartman et al. (2021) measured a mean $\delta^{34}S$ of 15.9 ‰ in the Netherlands, concluding that anthropogenic COS emissions in this region are small. Overall, it is expected that anthropogenic COS is likely to be more depleted than oceanic COS.

Experimental and theoretical studies have measured and calculated fractionation values for the different COS oxidation path-

ways. Hattori et al. (2012) measured a fractionation of –14.8 ‰ for the $CO^{34}S + O(^3P)$ reaction. Danielache et al. (2008) calculated a fractionation of –2.6 ‰ for the $CO^{34}S + OH$ reaction, whereas Schmidt et al. (2012) calculated a range between –5 to 0 ‰ for the same reaction. These negative fractionation values imply that the rate constant of the light isotopologue ($^{32}S$) is faster than the rate constant of the heavier isotopologue ($^{34}S$), making the remaining pool of COS enriched in $^{34}S$. Schmidt et al. (2013) modelled COS photolysis and concluded that there was a small and negative fractionation for $^{34}S$. Hattori et al. (2011)

measured a marginal fractionation for COS photolysis as well, while Lin et al. (2011) estimated a fractionation between –10.5 to 5.3 ‰. Recently, Yousefi et al. (2019) derived a positive fractionation of about +6 ‰ in the tropical stratosphere, based on satellite observations. Based on balloon measurements, Leung et al. (2002) and Colussi et al. (2004) suggested relatively large, positive isotope effects for COS photolysis with fractionations of $+73.8 \pm 8.6‰$ and $+67 \pm 7‰$, respectively and postulated that this would result in highly enriched sulfate in the stratosphere, leaving a pool of depleted COS.

Little information is available concerning fractionation during uptake by the biosphere. Using the binary diffusion theory Angert et al. (2019) calculated a fractionation of –5 ‰, implying an enrichment of atmospheric COS in $^{34}S$. Recent measurements on one plant species (Scindapsus Aureus branch) place it at –1.9 ‰ (Davidson et al., 2021).

Regarding the isotopic signature of SSA, the final product of COS photolysis, only one study exists that measured the corresponding isotopic signature of SSA $\delta^{34}S$. For non-volcanic SSA in the stratosphere, a value of $+2.6 \pm 0.3‰$ between 18–19km was reported by Castleman et al. (1974).

From the above considerations, it is clear that the COS budget and the corresponding sulfur isotopic budgets are still far

from being completely understood. In order to bridge this gap, we present a model study that aims to simulate the full atmospheric budget of sulfur, including its isotopes. By using available observations of concentrations and isotopic composition, we explore the following questions:

1. What is the COS contribution to SSA?

2. How can isotopic information help constrain the sulfur budget?





3. What are the largest uncertainties in the COS isotopic budget?

In this pioneering study, we use a 1-D column model, with the full atmospheric sulfur chemistry to explore the fate of sulfur in the atmosphere. We describe how the budgets are calculated, and study the isotope profiles of COS, $SO_2$ and sulfate in order to understand the profiles of the sulfur isotopic composition in the atmosphere. In Section 2 we discuss the methodology,

in particular the 1D model set up (Section 2.1) for the sulfur chemistry, including the sulfur isotopologues (Section 2.2). We also discuss how we calculate the atmospheric budget for COS, $SO_2$ and sulfate and the COS isotopic budget (Section 2.3), followed by a description of the sensitivity analyses we performed (Section 2.4). In Section 3, we present the aforementioned budgets and sensitivity analyses, followed by a discussion in Section 4 on the uncertainties in the model and the COS budget.

## 2 Methodology

In this section the set-up of the 1-dimensional column model is described. We explore how the model is equipped to simulate the full sulfur chemistry in the atmosphere. This includes a brief description on how the radiative transfer through the atmospheric column is modelled to obtain height-dependent photolysis frequencies. We then discuss the set-up that includes sulfur isotopologues as separate species, from which isotopic ratios of atmospheric sulfur are derived. The budget analysis of the sulfur S gases (COS, $SO_2$, and sulfate) in the atmosphere is then discussed, as is the budget calculation of the sulfur isotope

delta ($\delta^{34}$S) for these gases. Lastly, we describe the sensitivity analysis we carried out in our model, in order to encompass some of the uncertainties and unknowns of the COS isotopic budget.

### 2.1 Model description

We use the 1-dimensional column model PATMO (Planetary ATMOSpheres), which simulates the Earth's atmosphere up to 60 km. The model is based on the KROME code (Grassi et al., 2014) and the discretization follows Hu et al. (2012) and Rimmer

and Helling (2016). The code is presented in Ávila et al. (2021) and will be further explained in Danielache et al. (2022, in prep). At the surface, sulfur gases are emitted into the lowest 1 km grid cell of the model. To this end, emissions of COS, $CS_2$, $H_2S$, $SO_2$, $CH_3SCH_3$ are converted from teragrams of sulfur per year (Tg S yr$^{-1}$) to molecules cm$^{-3}$ s$^{-1}$, in the lowest model layer. COS and $CS_2$ emissions are obtained from Whelan et al. (2018), Zumkehr et al. (2017) and Suntharalingam et al. (2008). However, the COS budget as reported in the literature is not closed. We include therefore a missing source in our emissions

as well, in order to reach a steady state that matches the available observations. The emission rates for the other S gases are taken from Watts (2000), Khalil and Rasmussen (1984), Lee and Brimblecombe (2016) and Georgii and Warneck (1999). In the Supplement we provide a brief description of model performance, especially the mass balance, as well as a more extended model set up.

We include deposition of sulfur gases by dry and wet deposition and aerosol sedimentation. Dry deposition includes the uptake of COS by soils and the biosphere (Whelan et al., 2018), and takes place in the lowest model layer. Wet deposition represents the removal of S-gases by precipitation. Wet removal of species is restricted between the surface and 12 km. Details on the





wet removal calculation are given in the Supplement (Section S2). Dry deposition velocities and emissions of the different S-species are listed in Table 1.

**Table 1.** Emission rates and deposition velocities of the sulfur compounds. The last column gives the effective Henry's Law constants or solubility constants, in mol atmosphere[-1] (Giorgi and Chameides, 1985) that are used for the calculation of wet deposition velocities. The effective rainout lifetime is plotted in the Supplement (Figure S2).

| Species | Emission (Tg S yr[-1]) | Dry deposition (cm s[-1]) | Henry's constant (M atm[-1]) |
|---|---|---|---|
| COS | 0.62 | $8.8 \times 10^{-3}$ | 0.02 |
| $CS_2$ | 0.66 | $3 \times 10^{-4}$ | 0.05 |
| $H_2S$ | 9.4 | - | 0.1 |
| $SO_2$ | 50.5 | $7.15 \times 10^{-2}$ | $4 \times 10^3$ |
| $CH_3SCH_3$ | 22.6 | - | - |
| $SO_4$* (aerosol) | - | - | $5 \times 10^{14}$** |

* $SO_4$ has a gravitational deposition rate in all layers derived from Kasten (1968)

** Calculated from Turco et al. (1979)

The temperature and pressure profiles of the atmosphere are prescribed, as are the profiles of the other relevant atmospheric gases: the major atmospheric gases such as $N_2$, $O_2$, $H_2O$, $CO_2$ and CO. Also prescribed are the profiles of important oxidants

5  namely OH, $O_3$, $HO_2$ and $O^3P$, and data are taken from 1976 US standard profiles (Krueger and Minzner, 1976), and bench-marked against the present-day Earth atmosphere following Hu et al. (2012).

In every 1 km thick layer, the model resolves time-dependent bimolecular, termolecular, and bidirectional chemical reactions using the DLSODES solver for ordinary differential equations (Hindmarsh, 1983) of the form shown in Equation (1).

$$\frac{\partial n_{i,j}}{\partial t} = P_{i,j} - n_{i,j} \cdot L_{i,j} - \frac{\partial \phi_{i,j}}{\partial z}, \tag{1}$$

where $n_{i,j}$ is the number density (molecules cm[-3]) of species $i$ in layer $j$, P is the corresponding chemical production rate (molecules cm[-3] s[-1]), n·L is the chemical destruction rate (molecules cm[-3] s[-1]), and $\phi$ is the vertical transport flux (molecules cm[-2] s[-1]) (Hu et al., 2012). The vertical flux has a turbulent eddy diffusion component and a molecular diffusion component along the vertical (z-axis) (Hu et al., 2012). However, the molecular diffusion ($O$ 10[-3] cm[2] s[-1]) is much smaller than the eddy

15  diffusion in the 60 km of the atmosphere we simulate. Hence the vertical mixing is described as:

$$\phi_{i,j} = -K_j N_j \frac{\partial f_{i,j}}{\partial z}. \tag{2}$$

Here, $K_j$ is the turbulent eddy diffusion coefficient (cm[2] s[-1]), as presented in Supplementary Figure S1. The $N_j$ is the total density in layer j, and the $f_{i,j}$ is the mixing ratio = $n_{i,j}/N_j$. For K, empirical values from Massie and Hunten (1981) were used that range from 10[5] cm[2] s[-1] close to the surface and in the upper stratosphere to 10[3] cm[2] s[-1] in the lower stratosphere. The

20  default K values were multiplied by a factor of 2 to ensure that the stratospheric COS turnover (determined by transport and





stratospheric photolysis) amounts to about 40 Gg S yr$^{-1}$, as reported in the literature (Brühl et al., 2012)). Most of the COS transport to the stratosphere takes place in the tropics in the upward branch of the Brewer-Dobson circulation, and we expect that higher eddy diffusion coefficients are needed in our model to match the transport of COS to the stratosphere.

The full chemical reaction scheme and the photochemistry that is solved in PATMO are shown in Table 2, Table 3 and Table 4. Stratospheric sulfur chemistry is driven by photochemical reactions that depend on the amount of ultraviolet radiation that reaches each layer. Rather than the full radiative transfer, we only consider direct solar radiation that is attenuated by absorption of ultraviolet light, caused by species aloft. Since stratospheric photolysis is our main focus, we restrict the calculations to UV radiation only (Bian and Prather, 2002).

The solar flux (photons cm$^{-2}$ s$^{-1}$ nm$^{-1}$) is attenuated by absorption using the Beer-Lambert equation:

$$I_\lambda(z) = I_\lambda(\infty)e^{-\frac{\tau_z}{cos\theta}} \tag{3}$$

where $\theta$ is the solar zenith angle. $I_\lambda(\infty)$ is the spectral irradiance at the top of the atmosphere, which is attenuated by the overhead optical depth ($\tau_z$) that is calculated as:

$$\tau_{z,\lambda} = (z_{top} - z) \sum_i n_i \sigma_{i,\lambda} \tag{4}$$

($z_{top} - z$) represents the total thickness from the top of the atmosphere to altitude z (in cm). In our model the $z_{top}$ is 60 km. The $\tau$ takes into account the number density $n$ (molecules cm$^{-3}$) and wavelength-dependent, UV absorption cross-sections, $\sigma_\lambda$ (cm$^2$ molecule$^{-1}$) of molecule $i$. Subsequently, $\tau$ is summed over all the relevant species in the model (O2, O3 and all the S gases).

Using spectral information at height $z$, photolysis frequencies (s$^{-1}$) are calculated from the actinic flux $I_{(\lambda)}$, absorption cross section $\sigma$ and quantum yield q according to:

$$J_i(z) = \frac{1}{2} \cdot \int_{\lambda_1}^{\lambda_2} q_i(\lambda)\sigma_i(\lambda)I_\lambda(z)d\lambda, \tag{5}$$

We calculate the height-dependent photolysis, where the wavelength integral runs over the range 180–400 nm. This spectral range is relevant for the photo-dissociation reactions included in the model, namely all the sulfur species, O$_2$ and O$_3$. The entire
integral is multiplied by 1/2 to take the diurnal cycle into account and the solar zenith angle is set at 57.3° to represent the mean planetary angle (Hu et al., 2012). We solve the integral by numerical integration with a spectral resolution $d\lambda$ of 0.05 nm. For COS, the spectral resolution should not significantly change the isotopic effect due to its broad spectrum.

The photochemical transformations in the stratosphere lead to the formation of stratospheric sulfur aerosol (SSA) (Brühl
et al., 2012). Oxidation of COS leads, via SO$_2$, to gas-phase H$_2$SO$_4$, which nucleates easily to SSA under cold stratospheric conditions. Nucleation causes a phase change and a decrease in the Gibbs free energy as calculated by Hamill et al. (1977).



**Table 2.** Chemical Reaction Scheme. In the reaction rates, T is the absolute temperature in Kelvin. Second-order reaction rates in $cm^3$ molecule$^{-1}$ s$^{-1}$

| Number | Reaction | Reaction Rate | Reference |
|--------|----------|---------------|-----------|
| R1 | $COS + OH \longrightarrow CO_2 + SH$ | $1.1 \cdot 10^{-13} \cdot e^{\frac{-1200}{T}}$ | (Sander et al., 2011) |
| R2 | $COS + O \longrightarrow CO + SO$ | $2.1 \cdot 10^{-11} \cdot e^{\frac{-2200}{T}}$ | (Sander et al., 2011) |
| R3 | $CS_2 + OH \longrightarrow 1.17\,SH + 0.83\,COS^a$ | $\frac{1.25 \cdot 10^{-16} \cdot e^{\frac{4550}{T}}}{T + 1.81 \cdot 10^{-3} \cdot e^{\frac{3400}{T}}} \cdot \frac{P}{1013}$ | (DeMore et al., 1977) |
| R4 | $CS_2 + O \longrightarrow CS + SO$ | $3.20 \cdot 10^{-11} \cdot e^{\frac{-650}{T}}$ | (Sander et al., 2011) |
| R5 | $CS + NO_2 \longrightarrow COS + NO$ | $7.60 \cdot 10^{-17}$ | (Sander et al., 2011) |
| R6 | $CS + O_2 \longrightarrow COS + O$ | $2.9 \cdot 10^{-19}$ | (Sander et al., 2011) |
| R7 | $CS + O_3 \longrightarrow COS + O_2$ | $3.0 \cdot 10^{-16}$ | (Sander et al., 2011) |
| R8 | $CS + O \longrightarrow CO + S$ | $2.70 \cdot 10^{-10} \cdot e^{\frac{-761}{T}}$ | (Sander et al., 2011) |
| R9 | $H_2S + OH \longrightarrow H_2O + SH$ | $6.10 \cdot 10^{-12} \cdot e^{\frac{-75}{T}}$ | (Sander et al., 2011) |
| R10 | $H_2S + O \longrightarrow OH + SH$ | $9.22 \cdot 10^{-12} \cdot e^{\frac{-1803}{T}}$ | (Sander et al., 2011) |
| R11 | $H_2S + H \longrightarrow H_2 + SH$ | $8.00 \cdot 10^{-13}$ | (Wojciechowski et al., 1979) |
| R12 | $H_2S + HO_2 \longrightarrow H_2O + HSO$ | $3.00 \cdot 10^{-15}$ | (Sander et al., 2011) |
| R13 | $SH + O \longrightarrow H + SO$ | $1.60 \cdot 10^{-10}$ | (Sander et al., 2011) |
| R14 | $SH + O_2 \longrightarrow OH + SO$ | $4.00 \cdot 10^{-19}$ | (Sander et al., 2011) |
| R15 | $SH + O_3 \longrightarrow HSO + O_2$ | $9.00 \cdot 10^{-12} \cdot e^{\frac{-280}{T}}$ | (Sander et al., 2011) |
| R16 | $SH + NO_2 \longrightarrow HSO + NO$ | $3.00 \cdot 10^{-11} \cdot e^{\frac{250}{T}}$ | (Sander et al., 2011) |
| R17 | $SO + O_3 \longrightarrow SO_2 + O_2$ | $4.50 \cdot 10^{-12} \cdot e^{\frac{-1170}{T}}$ | (Atkinson et al., 2004) |
| R18 | $SO + O_2 \longrightarrow SO_2 + O$ | $1.60 \cdot 10^{-13} \cdot e^{\frac{-2282}{T}}$ | (Atkinson et al., 2004) |
| R19 | $SO + OH \longrightarrow SO_2 + H$ | $2.70 \cdot 10^{-11} \cdot e^{\frac{335}{T}}$ | (Sander et al., 2011) |
| R20 | $SO + NO_2 \longrightarrow SO_2 + NO$ | $1.40 \cdot 10^{-11}$ | (Sander et al., 2011) |
| R21 | $S + O_2 \longrightarrow SO + O$ | $2.31 \cdot 10^{-12}$ | (Sander et al., 2011) |
| R22 | $S + O_3 \longrightarrow O_2 + SO$ | $1.20 \cdot 10^{-11}$ | (Sander et al., 2011) |
| R23 | $S + OH \longrightarrow H + SO$ | $6.59 \cdot 10^{-11}$ | (Sander et al., 2011) |
| R24 | $SO_2 + HO_2 \longrightarrow OH + SO_3$ | $1.00 \cdot 10^{-18}$ | (Sander et al., 2011) |
| R25 | $SO_2 + NO_2 \longrightarrow SO_3 + NO$ | $2.32 \cdot 10^{-26}$ | (Sander et al., 2011) |
| R26 | $SO_2 + O_3 \longrightarrow SO_3 + O_2$ | $3.00 \cdot 10^{-12} \cdot e^{\frac{-7000}{T}}$ | (Sander et al., 2011) |
| R27 | $HSO + O_2 \longrightarrow SO_2 + OH$ | $1.69 \cdot 10^{-15}$ | (Bulatov et al., 1986) |
| R28 | $HSO + O_3 \longrightarrow 2\,O_2 + SH$ | $2.54 \cdot 10^{-13} \cdot e^{\frac{-392.4}{T}}$ | (Wang and Howard, 1990) |
| R29 | $HSO + NO_2 \longrightarrow NO + HSO_2$ | $9.60 \cdot 10^{-12}$ | (Sander et al., 2011) |
| R30 | $HSO_2 + O_2 \longrightarrow HO_2 + SO_2$ | $3.01 \cdot 10^{-13}$ | (Sander et al., 2011) |
| R31 | $HSO_3 + O_2 \longrightarrow HO_2 + SO_3$ | $1.30 \cdot 10^{-12} \cdot e^{\frac{-330}{T}}$ | (Sander et al., 2011) |
| R32 | $SO_2 + O + M \longrightarrow SO_3 + M$ | * | (Sander et al., 2011) |
| R33 | $SO_2 + OH + M \longrightarrow HSO_3 + M$ | ** | (Sander et al., 2011) |
| R34 | $SO_3 + H_2O \longrightarrow H_2SO_4$ | $1.20 \cdot 10^{-15}$ | (Reiner and Arnold, 1994) |
| R35 | $CH_3SCH_3 + O \longrightarrow 0.99\,SO_2 + 0.007\,COS^b$ | $1.0 \cdot 10^{-11} \cdot e^{\frac{410}{T}}$ | (DeMore et al., 1977) |
| R36 | $CH_3SCH_3 + OH \longrightarrow SO_2 + 0.007\,COS^b$ | $1.2 \cdot 10^{-11} \cdot e^{\frac{-260}{T}}$ | (DeMore et al., 1977) |
| R37 | $CH_3SCH_3 + OH \longrightarrow 0.74\,SO_2 + 0.007\,COS^b + 0.25\,MSA$ | *** | (Chatfield and Crutzen, 1990) |

[a] 0.83 yield from Stickel et al. (1993)

[b] 0.007 yield from Barnes et al. (1994, 1996); Albu et al. (2006)

\* $k_o = 1.80 \cdot 10^{-33} \cdot \frac{T}{300}^{2}$ ; $k_\infty = 4.20 \cdot 10^{-14} \cdot \frac{T}{300}^{1.8}$

\** $k_o = 3.30 \cdot 10^{-31} \cdot \frac{T}{300}^{-4.3}$ ; $k_\infty = 1.60 \cdot 10^{-12}$

\*** $3.04 \cdot 10^{-12} \cdot e^{\frac{350}{T}} \cdot \frac{\gamma}{1+\gamma}$ ; $\gamma = 5.53 \cdot 10^{-31} \cdot e^{\frac{7460}{T}} \cdot [O_2]$





**Table 3.** Photochemical reactions considered in the model. As the model calculates the rates convolving the cross-sections, the actinic flux and the quantum yield, the table shows the wavelength dependent cross sections (cm$^2$ molecules$^{-1}$) considered, and the wavelength regions where the quantum yield (molecules photons$^{-1}$) of the reaction is 1. The wavelength range described as No data is set to zero because there is no data on the absorption cross section.

| Number | Reaction | Quantum Yield = 1 | Cross-sections |
|---|---|---|---|
| R38 | $O_2 + h\nu \longrightarrow O + O$ | $\lambda < 240$ nm | $\lambda = 180\text{-}181$ nm (Kockarts, 1976) |
| | | | $\lambda = 181\text{-}235$ nm (Ogawa, 1971) |
| | | | $\lambda = 235\text{-}400$ nm (Bogumil et al., 2003) |
| R39 | $O_3 + h\nu \longrightarrow O_2 + O$ | | $\lambda = 180\text{-}230$ nm (Sander et al., 2011) |
| | | | $\lambda = 230\text{-}400$ nm (Malicet et al., 1995) |
| R40 | $COS + h\nu \longrightarrow CO + S$ | $\lambda < 285$ nm | $\lambda = 180\text{-}185$ nm No data |
| | | | $\lambda = 185\text{-}195$ nm (Limão-Vieira et al., 2015) |
| | | | $\lambda = 195\text{-}260$ nm (Hattori et al., 2011) |
| | | | $\lambda = 260\text{-}300$ nm (Limão-Vieira et al., 2015) |
| | | | $\lambda = 300\text{-}400$ nm No data |
| R41 | $CS_2 + h\nu \longrightarrow CS + S$ | $\lambda < 272$ nm | $\lambda = 180\text{-}194$ nm (Chen and Robert Wu, 1995) |
| | | | $\lambda = 194\text{-}205$ nm (Sunanda et al., 2015) |
| | | | $\lambda = 205\text{-}275$ nm (Grosch et al., 2015) |
| | | | $\lambda = 275\text{-}370$ nm (Sander et al., 2011) |
| | | | $\lambda = 370\text{-}400$ nm No data |
| R42 | $H_2S + h\nu \longrightarrow SH + H$ | $\lambda < 313$ nm | $\lambda = 180\text{-}260$ nm (Wu and Chen, 1998) |
| | | | $\lambda = 260\text{-}370$ nm (Grosch et al., 2015) |
| | | | $\lambda = 370\text{-}400$ nm No data |
| R43 | $SO + h\nu \longrightarrow S + O$ | | $\lambda = 180\text{-}260$ nm (Danielache et al., 2014) |
| | | | $\lambda = 260\text{-}400$ nm No data |
| R44 | $SO_2 + h\nu \longrightarrow SO + O$ | $\lambda < 217$ nm | $\lambda = 180\text{-}189.5$ nm (Danielache et al., 2008) |
| | | | $\lambda = 189.5\text{-}225$ nm (Endo et al., 2015) |
| | | | $\lambda = 225\text{-}239$ nm (Wu et al., 2000) |
| | $SO_2 + h\nu \longrightarrow {}^1SO_2$ | $240$ nm $< \lambda < 320$ nm | $\lambda = 239\text{-}250$ nm (Bogumil et al., 2003) |
| | | | $\lambda = 250\text{-}320$ nm (Danielache et al., 2014) |
| | $SO_2 + h\nu \longrightarrow {}^3SO_2$ | $320$ nm $< \lambda < 400$ nm | $320\text{-}400$ nm (Bogumil et al., 2003) |
| R45 | $SO_3 + h\nu \longrightarrow SO_2 + O$ | $\lambda < 343$ nm | $\lambda = 180\text{-}330$ nm (Sander et al., 2011) |
| | | | $\lambda = 330\text{-}400$ nm No data |
| R46 | $H_2SO_4 + h\nu \longrightarrow SO_2 + OH + OH*$ | $\lambda < 234$ nm | $\lambda\ 120\text{-}130$ nm (Cheng et al., 2002) |
| | | | $\lambda = 135\text{-}230$ nm (Sander et al., 2011) |
| | | | $\lambda = 230\text{-}247$ nm (Brion et al., 2005) |
| | | | $\lambda = 247\text{-}400$ nm No data |

*Here HCl absorption cross sections and an approximation factor of 0.011 are used (Turco et al., 1979; Mills et al., 1999; Burkholder et al., 2000), since $H_2SO_4$ absorption cross sections are difficult to determine experimentally





**Table 4.** Photoexcitation Reactions considered in the model.

| Number | Reaction | Reaction Rate | Reference |
|---|---|---|---|
| R47 | $^1SO_2 + {}^1SO_2 \longrightarrow SO_3 + SO$ | $4.0 \cdot 10^{-12}$ | Turco et al. (1982) |
| R48 | $^3SO_2 + O_2 \longrightarrow SO_3 + O$ | $1.0 \cdot 10^{-16}$ | Turco et al. (1982) |
| R49 | $^3SO_2 + SO_2 \longrightarrow SO_3 + SO$ | $7.0 \cdot 10^{-14}$ | Turco et al. (1982) |
| R50 | $^1SO_2 \longrightarrow SO_2 + h\nu$ | $2.2 \cdot 10^{4}$ | Turco et al. (1982) |
| R51 | $^3SO_2 \longrightarrow SO_2 + h\nu$ | $1.1 \cdot 10^{3}$ | Turco et al. (1982) |
| R52 | $^1SO_2 \longrightarrow {}^3SO_2$ | $1.5 \cdot 10^{3}$ | Turco et al. (1982) |
| R53 | $^1SO_2 + M \longrightarrow SO_2 + M$ | $1.0 \cdot 10^{-11}$ | Turco et al. (1982) |
| R54 | $^1SO_2 + M \longrightarrow {}^3SO_2 + M$ | $1.0 \cdot 10^{-12}$ | Turco et al. (1982) |
| R55 | $^3SO_2 + M \longrightarrow SO_2 + M$ | $1.4 \cdot 10^{-13}$ | Turco et al. (1982) |

The phase change depends on temperature, number density, and the ratio of the partial pressure of the $H_2SO_4$ (gas), $P_a$, and the saturation vapour pressure of $H_2SO_4$, $P_v$. Hence, if $\frac{P_a}{P_v} > 1$, nucleation occurs and gas-phase $H_2SO_4$ condenses to liquid phase aerosol. We use the vapour phase and gas phase calculations from Hamill et al. (1977) to represent $P_v$ and $P_a$ in the stratosphere. In the model, the amount of sulfate that falls between $P_v$ (lower limit) and $P_a$ (upper limit) condenses into sulfate

(liquid aerosol-phase). Temperature and pressure dependent evaporation of aerosol is accounted for (Hamill et al., 1977), but not discussed further in this study. Concerning the aerosol processes, aerosols are removed by gravitational settling from the stratosphere, assuming an aerosol particle mean radius of 0.3 $\mu$m with a mass density of 1.0 g cm$^{-3}$. Below 12 km, wet removal becomes the main removal mechanism of sulfate aerosol.

Our aim is to analyse the steady state solution, in which sources, sinks, and transport of S-compounds are in equilibrium. To this end, we simulate 60 years and check if the simulation has converged (Supplement Figure S5). We find that the slowest timescale is in the stratosphere, where transport is slow, and photolysis frequencies depend critically on the overhead burdens of UV-absorbing S-gases.

## 2.2 Sulfur isotopologues

Our simulations include different isotopologues of all the sulfur gases as separate molecules. Sulfur has 4 stable isotopes ($^{32/33/34/36}S$). Isotopic ratios are usually reported relative to an international standard ratio:

$$\delta^x S = \frac{{}^x R_{\text{sample}} - {}^x R_{\text{standard}}}{{}^x R_{\text{standard}}} \cdot 1000\text{‰},\tag{6}$$

where x stands for 32, 33, 34 and 36, and $^x$R stands for the isotopic ratios ($^{32/33/34/36}S$) of samples and standards; sulfur isotope ratios are reported relative to the Vienna Canyon Diablo Triolite (VCDT) in permil (‰).

We tested the model to make sure we are not creating artificial isotopic effects. A zero fractionation test was therefore per-





formed. In this test the isotopologues of S are emitted per natural abundance (VCDT) and all the reaction rates are assumed the same for each isotope. We found that the numerical noise of the model is negligible and hence we conclude that the model is suitable for modelling isotopologues (Supplementary Figure S3 and discussed further in Danielache et al. (2022, in preparation)).

Since most measurements are available for $^{32}$S and $^{34}$S, we will concentrate on these isotopologues and the corresponding $\delta^{34}$S values. $^{32}$S and $^{34}$S have natural abundances of 95.02% and 4.25% respectively. In the model, sulfur gases are emitted to the atmosphere with an assumed $\delta^{34}$S source signature. We prescribe some of the reported source signatures to the emissions of $^{32}$S and $^{34}$S, respectively. These signatures are listed in Table 5.

**Table 5.** Source signatures of emitted sulfur gases

| Species | $\delta^{34}$S effective source signature (‰) | Reference |
| --- | --- | --- |
| COS | 10.5 | Davidson et al. (2021) |
| CS$_2$ | 10.4 | Davidson et al. (2021) |
| H$_2$S | 1.0 | Krouse and Grinenko (1991) |
| SO$_2$ | 5.0 | Mukai et al. (2001) |
| CH$_3$SCH$_3$ | 20.0 | Davidson et al. (2021) |

For the isotopic signature of COS emission we use 10.5 ‰, an effective emission signal derived from Davidson et al. (2021). This study reports anthropogenic COS to have a signal close to 8 ‰ and direct oceanic COS to be closer to 13 ‰. Although the contribution of anthropogenic versus marine sources is debated, we consider a 50-50 split as derived in Angert et al. (2019).

Davidson et al. (2021) also measured oceanic CS$_2$ isotopic signatures to be around 16 ‰, but the anthropogenic signature is unknown. With the assumption that the anthropogenic signature of CS$_2$ is the same as that of COS (8 ‰), and assuming that 70% of the CS$_2$ emission is anthropogenic (Angert et al., 2019), we use an effective emission signature of 10.4 ‰ for CS$_2$ (Chin and Davis, 1995). Lastly, Davidson et al. (2021) measured oceanic CH$_3$SCH$_3$ with a signal of 20 ‰, which we use in our model.

Chemical, physical, and biological processes may fractionate, which implies that a process proceeds sightly faster or slower for one isotopologue compared to the other. In this paper, we only consider fractionations for $^{34}$S. Fractionation often arises due to mass differences, and symbols $\alpha$ and $\epsilon$ are used to describe differences in rate coefficients $k$:

$$^{34}\alpha = \frac{^{34}k}{^{32}k} \quad ; \quad ^{34}\epsilon = {}^{34}\alpha - 1 \tag{7}$$

Values of $\epsilon$ are often reported in ‰. A negative value of $\epsilon$ implies that the light isotopologue reacts faster. The dominant

sink of COS is uptake by the biosphere through the stomata of leaves (Berry et al., 2013). We impose an $\epsilon_{bio}$ of –1.9 ‰ for the dry deposition process as measured by Davidson et al. (2021), implying that the lighter isotopologue diffuses slightly faster





**Table 6.** Isotopic Chemical Reaction Scheme with fractionation factors $\alpha$ included for $^{34}$S. In the reaction rate formulas, T is the absolute temperature in Kelvin.

| Number | Reaction | Reaction Rate | Fractionation factor ($\alpha$) |
|---|---|---|---|
| R1a | $CO^{32}S + OH \longrightarrow CO_2 + {}^{32}SH$ | $1.1 \cdot 10^{-13} \cdot e^{\frac{-1200}{T}}$ | |
| R1b | $CO^{34}S + OH \longrightarrow CO_2 + {}^{34}SH$ | $(= k_{1a} \cdot {}^{34}\alpha_1)$ | $^{34}\alpha_1 = 0.9967$ [1] |
| R2a | $CO^{32}S + O \longrightarrow CO + {}^{32}SO$ | $2.1 \cdot 10^{-11} \cdot e^{\frac{-2200}{T}}$ | |
| R2b | $CO^{34}S + O \longrightarrow CO + {}^{34}SO$ | $(= k_{2a} \cdot {}^{34}\alpha_2)$ | $^{34}\alpha_2 = 0.9783$ [2] |
| | | $k_o = 3.30 \cdot 10^{-31} \cdot \frac{T}{300}^{-4.3}$ | |
| R32a | $^{32}SO_2 + OH + M \longrightarrow H^{32}SO_3 + M$ | $k_\infty = 1.60 \cdot 10^{-12}$ | |
| R32b | $^{34}SO_2 + OH + M \longrightarrow H^{34}SO_3 + M$ | $(= k_{32a} \cdot {}^{34}\alpha_{32})$ | $^{34}\alpha_{32} = 1.0089$ [3] |

[1] at 18 km, these are in fact height-dependent, see Supplementary Figure S3 (Schmidt et al., 2012)

[2] (Hattori et al., 2012)

[3] (Harris et al., 2012)

and is therefore taken up preferentially by plants. Chemical reactions may also fractionate. We include fractionations for the following reactions: COS + O$^3$P (Hattori et al., 2012), COS + OH (Schmidt et al., 2012) and SO$_2$ + OH as in Table 6. The fractionation for COS + OH (Schmidt et al., 2012) is height-dependent. For COS + O($^3$P) (Hattori et al., 2012), fractionation is height independent. Note that the SO$_2$ + OH reaction has a positive $^{34}\epsilon$ (Harris et al., 2012). Not much is known about the fractionation that is associated with CS$_2$ and CH$_3$SCH$_3$ oxidation, therefore we currently assume no fractionation.

The photolysis frequencies are calculated using Equation (5). For COS, we use the cross-sections from Hattori et al. (2011), resulting in height-dependent $\epsilon$ as presented in the Supplementary Figure S4. Values reported in the literature vary considerably, ranging from +73 ‰ to −10.5 ‰ (Leung et al., 2002; Lin et al., 2011). Most studies expect this $\epsilon$ to be small, but whether the values are negative or positive is still debated (Schmidt et al., 2012; Yousefi et al., 2019).

## 2.3 Budget calculation

As we run the model to steady state, the change in concentration over time is below a threshold value at the end of the simulation, i.e. the sources and sinks are in balance with the imposed transport and radiation processes. Using the steady state assumption, the steady state mass balance equation for COS, SO$_2$ and sulfate (H$_2$SO$_4$ and SO$_4$) in the atmosphere is calculated as:

$$\frac{dC}{dt} = 0 = \text{Emission} + \text{Chemical Production} - \text{Chemical Loss} - \text{Transport} - \text{Deposition,} \tag{8}$$

in which chemical loss, transport and deposition are defined as negative contributions, and $C$ is the amount of S-compound in each layer, or integrated over the troposphere, stratosphere, or global atmosphere (i.e. the budget equation is in steady state in each layer). To separate the troposphere from the stratosphere, we define two boxes. Although the wet removal in the troposphere ends at 12 km, we consider the 12–16 region (Upper Troposphere/Lower Stratosphere (UTLS)) a part of the troposphere





due to enhanced mixing (See Supplementary Figure S1). Hence, for the budget, we define the troposphere as the sum of the lower 16 boxes (0–16 km), and the stratosphere as the remainder (16–60 km). The flux contributions are calculated in Gg S yr$^{-1}$ or Tg S yr$^{-1}$.

The same steady state approximation is used to derive a $\delta^{34}$S budget. This assumes, as before, that the $\delta^{34}$S value of all compounds in all layers has reached steady state. We develop the following isotopic $\delta$ budget equation:

$$\frac{d}{dt}\delta_A = 0 = \overbrace{(\delta_E - \delta_A)\frac{{}^{32}E}{{}^{32}C_A}}^{\text{Emission}} + \overbrace{(\alpha_y\alpha_k(\delta_{pre}+1) - (\delta_A+1))\frac{{}^{32}P}{{}^{32}C_A}}^{\text{Chemical Production}} + \overbrace{(\delta_A+1){}^{32}L(1-\alpha_l)}^{\text{First Order Loss}} + \overbrace{k_t(\delta_n - \delta_A)\frac{{}^{32}C_n}{{}^{32}C_A}}^{\text{Transport}}. \tag{9}$$

The step-by-step derivation of this equation is given in Supplement S3. In this equation, $\delta$ and $\alpha$ values are given as absolute values (Equation (7)). In the chemical production term, $\alpha_y$ represents the possible fractionation due to a yield difference (See

Supplement S3.3), while $\alpha_k$ and $\alpha_l$ are the ratios of the rate constants (Equation (7)). $\delta_A$ is the atmospheric signature of the chemical compound, $\delta_E$ is the emission signature, and $\delta_{pre}$ is the signature of the precursor S molecule, $\delta_n$ is the signature of the compound in a neighbouring layer. $^{32}$E is the amount emitted of the more abundant isotope (Tg S yr$^{-1}$). $^{32}C_A$ is the atmospheric abundance of the compound in question in the layer. $^{32}$P is the amount produced (in Tg S yr$^{-1}$) of the more abundant isotope. $^{32}C_n$ is the amount of the compound in a neighbouring layer (Tg S). $^{32}$L is the loss rate for the abundant

isotope (yr$^{-1}$). $k_t$ is a transport timescale in yr$^{-1}$ (see Supplement S3) . We present the $\delta$ budget contributions in ‰ yr$^{-1}$. A similar concept was used in Tans et al. (1993) and Van Der Velde et al. (2018) for carbon isotopes.

### 2.4    Sensitivity analyses

To assess the uncertainties in our calculations, we perform a number of sensitivity studies. There is relatively little information from isotope measurements, hence we want to assess how sensitive our simulations are for some key model parameters.

In our "BASE" simulation, photolysis frequencies of COS are based on radiative transfer calculations, which results in a negative fractionation (see Supplementary Figure S4), but with some dependence on altitude. To test the influence of the uncertain cross sections, we include sensitivity simulations with prescribed fractionation. In the literature, the photolysis fractionation has been reported as both positive (which would result in isotopically lighter COS in the stratosphere) or negative (vice versa).

Hence, we include simulations where we impose a positive $\epsilon$(J$_{\text{COS}}$ + xx ‰) for the COS photolysis reaction and a negative $\epsilon$ (J$_{\text{COS}}$ −yy ‰) to study the subsequent effect on the isotopic signature of the sulfur compounds in the stratosphere. The positive $\epsilon$ = +73 ‰ from Leung et al. (2002) was derived from balloon measurements. Lin et al. (2011) calculated a range of epsilons and we utilise −10.5 ‰, to take into account the negative $\epsilon$ value for COS photolysis. We also include a run with no photolysis fractionation.

The next sensitivity tests the impact of a possible fractionation in the CS$_2$ + OH reaction, as this is an important source of





atmospheric COS. $CS_2$ contains 2 S atoms, and currently it is unclear how the $\delta^{34}$S signature of $CS_2$ will propagate to COS (Zeng et al., 2017). We also impose a yield of 0.83 towards COS on this reaction based on Stickel et al. (1993). The fast reaction of $CS_2 + OH$ completely converts $CS_2$ to COS and a $SO_2$ precursor. Hence, in order to account for a yield fractionation the yield needs to be adjusted just for the $^{34}S$ reaction. We impose more (less) $^{34}S$ towards COS and in turn making atmospheric

COS heavier (lighter). We include a yield fractionation of 6 ‰, and we implement this fractionation by shifting the yield to either 0.835 or 0.825 towards COS and 1.165 or 1.175, respectively towards the $SO_2$ precursor, (see R3 Table 2).

In order to understand the significance of COS for $SO_2$ and SSA in the model, we also include a simulation without COS. With this we study the effect of COS on the $SO_2$ and sulfate signatures, especially in the stratosphere, and highlight the impor-
tance of COS for SSA formation.

The BASE $\delta$ value of COS emission from Davidson et al. (2021) is 10.5 ‰, which is an effective emission signal, resulting from a combination of anthropogenic and marine emissions. The expected anthropogenic signal inferred from the measurements was 8 ‰, while the marine signal is expected to be about 14 ‰. In an effort to show what the final signal would be
if we only had purely anthropogenic versus purely marine emissions, we carry out simulations ($E_{COS}$) between the different COS emission signatures: $\delta^{34}$S = 8 ‰ (Davidson et al., 2021) and $\delta^{34}$S = 14 ‰ (Hattori et al., 2020). This sensitivity also encapsulates the range of current atmospheric measurements for COS in the atmosphere (Kamezaki et al., 2019; Hattori et al., 2020; Baartman et al., 2021), and shows the possible range of what the expected COS atmospheric signature might be, and subsequently the $SO_2$ and sulfate isotopic signature in the stratosphere.

The BASE biosphere fractionation from Davidson et al. (2021) was taken as –1.9 ‰. In order to study the importance of this biosphere fractionation (BIO), we include a simulation with no biosphere fractionation ($\epsilon_{BIO}$ = 0 ‰), and another with biosphere fractionation of –5 ‰, as calculated by Angert et al. (2019).

## 3   Results

### 3.1   Vertical profiles

Figure 1 shows the steady state vertical profiles for COS, $SO_2$ and sulfate, resulting from the BASE simulation. In the top panels the mixing ratios are plotted. For COS, the tropospheric mixing ratio amounts to 527 ppt, close to the 500 ppt observed for tropospheric COS (Montzka et al., 2007). In the stratosphere, the COS profile decays rapidly due to photolysis, and at 40 km altitude there is almost no COS left. In the $SO_2$ profile, there is a significant decrease in the troposphere with height, as $SO_2$
oxidation and wet deposition are prominent in the troposphere. Above the $SO_2$ minimum around 20 km, a slight enhancement towards the top of the atmosphere is observed (note the logarithmic scale). COS is the major source of $SO_2$ in this region, but $SO_2$ is quickly oxidized to sulfate. Higher up in the stratosphere, sulfate photolyses back to $SO_2$, hence there is an equilibrium between $SO_2$ and sulfate. For sulfate that is produced from $SO_2$, we clearly model a Junge layer (as labelled) in the stratosphere,





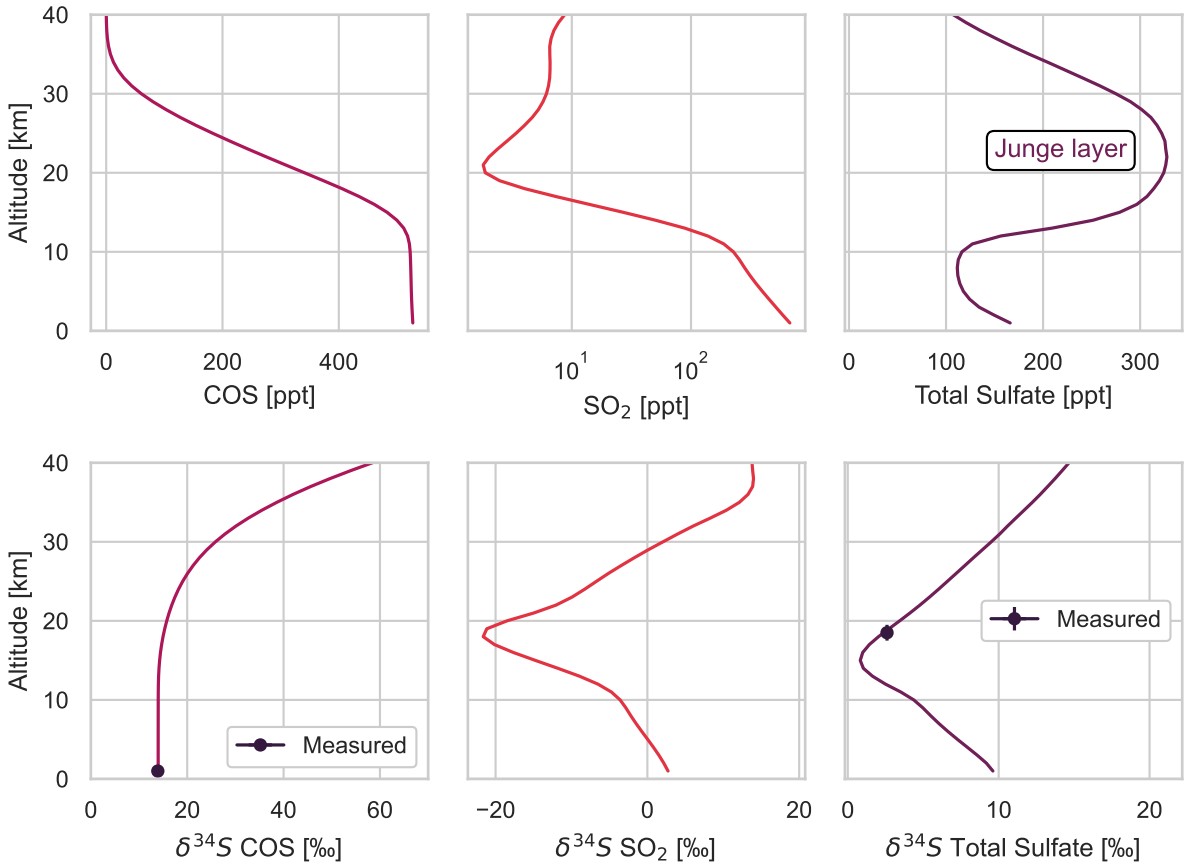

**Figure 1.** Vertical profile of carbonyl sulfide (COS), sulfur dioxide (SO$_2$) and sulfate up to 40 kilometers in the atmosphere. The top panel shows the mixing ratios of the three molecules in parts per trillion (ppt). The SO$_2$ mixing ratio is plotted with a logarithmic scale. The bottom panel shows the $\delta^{34}$S profiles for the three compounds in ‰. The provided $\delta^{34}$S measurements are 13.9 ‰ for COS (Davidson et al., 2021) and 2.6 ‰ for stratospheric sulfate (Castleman et al., 1974)

with a peak of about 320 ppt at 20-25 km.

The lower panels in Figure 1 show the simulated $\delta^{34}$S profiles in steady state. The COS isotopic signal is about 14 ‰ in the troposphere, very close to the measured 13.9 ‰ reported by Davidson et al. (2021), and within the range measured by Hattori et al. (2020). In the stratosphere, COS gets increasingly enriched in $^{34}$S with height. This is due to COS removal by photolysis. The prescribed negative $^{34}\epsilon$, which removes the lighter COS faster, leads to a continuous isotope enrichment of COS in the stratosphere. The simulated $\delta^{34}S$ value reaches 58 ‰ at 40 km, though at this height there is hardly any COS left in the stratosphere. For SO$_2$ $\delta^{34}$S = 3 ‰ at the surface, reflecting the source signature of the emitted SO$_2$ (Table 5). As we





continue up the troposphere, the $SO_2$ isotopic signature is increasingly depleted, largely due to the inverse kinetic effect during OH oxidation (Table 6). As a result, the heavier isotope reacts away faster, leaving a pool of lighter $SO_2$ in the troposphere. At 17 km, the $SO_2$ signature reaches a value of almost –22 ‰, however very little $SO_2$ is found at this height; it is converted to sulfate. In the stratosphere, COS becomes the major source of $SO_2$. As a result, a progressive enrichment of $SO_2$ is observed

towards 40 km, reflecting the isotopically enriched COS precursor. Similarly, the sulfate isotopic signature largely follows the signal of $SO_2$. Sulfate is enriched with respect to its precursor $SO_2$, due to heavier S reacting favourably away from $SO_2$ in the OH oxidation reaction ($\epsilon$ = +8.9 ‰, see Table 6). Once COS photolysis becomes significant, the sulfate signal also becomes more isotopically enriched. Between 18–19 km, the modelled sulfate signal falls between 2–3 ‰, matching very well with the $2.6 \pm 0.3$ ‰ measured by Castleman et al. (1974). However, it is important to note that the sulfate $\delta^{34}S$ in the stratosphere has

a strong vertical gradient, which also vary with latitude. At about 40 km the signal is strongly enriched compared to 20 km altitude, with a $\delta^{34}S$ of almost 15 ‰.

It is interesting to observe that all the modelled sulfur compounds get enriched with height in the stratosphere. In the Supplement Section 4 we present the total S budget in the atmosphere. Total S gets increasingly enriched in the stratosphere due

to to the gravitational settling term of sulfate, which is the main sink of S in the stratosphere (Supplementary figure S9). Gravitational settling removes S that is relatively light compared to the signature of total S. $SO_4$ is $\approx 10‰$ at 30 km (Figure 1) compared to $\approx 15‰$ for total S at 30 km (Supplementary figure S6). The positive tendency in the $\delta$ budget (Equation (9)) is compensated by a negative tendency of transport to bring lighter COS to the stratosphere.

### 3.2 COS Budget

**Table 7.** Carbonyl sulfide (COS) budget in the troposphere (below 16 km) and stratosphere (above 16 km) in Gg S yr$^{-1}$

| Process | Tropospheric flux | Stratospheric flux |
|---|---:|---:|
| Emission | **+ 618.3** | 0 |
| Chemical production (total) | **+ 436.6** | 0 |
|    from $CS_2$ | 278.6 | 0 |
|    from $CH_3SCH_3$ | 158 | 0 |
| Chemical loss | **– 84.1** | **– 40** |
| Transport | **– 40** | **+ 40** |
| Dry Deposition | **– 930.2** | 0 |

In Table 7, we present the tropospheric and stratospheric budgets of COS. 618.3 GgS yr$^{-1}$ of COS is emitted at the surface in our model. This value includes direct sources of COS from oceans, anthropogenic, biomass burning and also includes an unidentified flux of 276 Gg S yr$^{-1}$. This gap in the COS budget is smaller compared to Berry et al. (2013) mainly due to updated anthropogenic emissions (Campbell et al., 2015; Zumkehr et al., 2017; Whelan et al., 2018; Ma et al., 2021).





The total chemical production amounts to 436.6 GgS yr$^{-1}$, and is a combination of $CS_2$ and $CH_3SCH_3$ oxidation. As we utilise the 83% yield from Stickel et al. (1993), we get about 278.6 GgS yr$^{-1}$ from $CS_2$ oxidation. From $CH_3SCH_3$ oxidation we prescribe a contribution of 158 GgS yr$^{-1}$ of COS, similar to Ma et al. (2021). We note here that $CH_3SCH_3$ oxidation to COS is considered feasible only in very pristine, low NOx conditions. However, as marine $CH_3SCH_3$ emissions are very large, we
assume some COS formation, albeit with a very small yield of 0.7% (Barnes et al., 1994, 1996; Albu et al., 2006).

The largest COS sink is uptake by the biosphere, which removes about 930.2 Gg S yr$^{-1}$ in our model, in line with Berry et al. (2013). Note the deposition rate was adjusted such that the COS mixing ratio (527 ppt in the model) matches atmospheric observations (Kettle, 2002; Montzka et al., 2007). COS oxidation (mainly by OH) to $SO_2$ accounts for 84.1 Gg S yr$^{-1}$. This
number is on the lower end of other studies that have modeled the OH sink to be between 82-116 Gg S yr$^{-1}$ (Kettle, 2002; Montzka et al., 2007; Ma et al., 2021).

The tropospheric lifetime, which is the tropospheric burden divided by the removal rate of COS (dry deposition, oxidation and net transport to the stratosphere), in our model is 2.2 years, which lines up well with the 2–3 years calculated in literature
(Brühl et al., 2012).

Due to the long lifetime of COS in the troposphere, a net amount of 40 Gg S yr$^{-1}$ is moved to the stratosphere comparable with the estimate by Sheng et al. (2015) of 40.7 Gg S yr$^{-1}$. In the stratosphere, COS is photolysed to $SO_2$ and then further converted to SSA. The calculated stratospheric burden of COS is 130 Gg S, which is only 5.4% of the tropospheric burden.

We calculate the stratospheric lifetime, defined as the stratospheric burden of COS divided by the stratospheric chemical destruction, to be about 3.3 years. The lifetime defined as total atmospheric burden divided by stratospheric loss amounts to 60.2 years, consistent with literature estimates (Sheng et al., 2015; Brühl et al., 2012)

In the next section we explore the full stratospheric sulfur budget, to investigate the importance of COS for SSA formation.

### 3.3 Stratospheric Sulfur Budget

**Table 8.** Sulfur stratospheric budget (above 16 km) in Gg S yr$^{-1}$ for COS, $SO_2$, and sulfate.

| Process | COS flux | $SO_2$ flux | Sulfate flux |
|---|---|---|---|
| Chemical production | 0 | + 40 | + 52 |
| Chemical loss | − 40 | − 52 | 0 |
| Transport | + 40 | + 12 | − 13 |
| Gravitational settling | 0 | 0 | − 39 |





Table 8 shows the stratospheric sulfur budget for COS, $SO_2$ and sulfate. As we saw in the previous section, about 40 GgS yr$^{-1}$ of COS enters the stratosphere, which is oxidised to $SO_2$ and sulfate. About 12 GgS yr$^{-1}$ of $SO_2$ also enters the stratosphere which is also transferred to sulfate. Hence, about 52 GgS yr$^{-1}$ of sulfate is produced in the stratosphere. COS thus accounts for 77% of the SSA formation.

In the troposphere, sulfate is quickly removed due to efficient wet removal. As the eddy transport in the model depends on the gradient, the removal of sulfate in the troposphere and production of sulfate in the stratosphere leads to a sulfate (eddy-driven) transport of 13 Gg S yr$^{-1}$ towards the troposphere. An additional 39 GgS yr$^{-1}$ is lost to the troposphere due to gravitational settling of sulfate aerosol; this is related to the gravitational deposition rate which is prescribed per layer. For completeness, we

provide the tropospheric budgets of $SO_2$ and sulfate in the Supplement (Supplementary Table 1). We exclude the quick cycling between sulfate and $SO_2$ that occurs in the upper atmosphere (> 40 km, (Brühl et al., 2012)), since these processes have little impact on the COS budget.

### 3.4   Isotopic Budget of COS

**Table 9.** Carbonyl sulfide (COS) $\delta^{34}$S budget in the troposphere (below 16 km) and stratosphere (above 16 km) in permil per year (‰ yr$^{-1}$). Negative numbers imply that the process depletes atmospheric COS, whereas positive numbers imply enrichment of atmospheric COS.

| Process | Tropospheric flux | Stratospheric flux |
|---|---|---|
| Emission | − 0.95 | 0 |
| Chemical production | − 0.02 | 0 |
| Chemical loss | + 0.11 | + 1.96 |
| Transport | + 0.07 | − 1.96 |
| Dry Deposition | + 0.79 | 0 |

Table 9 shows the isotopic contributions to the atmospheric signal of COS $\delta^{34}$S, calculated according to Equation (9). We

found that a 60 year simulation is sufficient to reach steady state for the isotopes as well (See Supplement, Figure S6). In the troposphere, the processes of emission and chemical production deplete the atmospheric CO$^{34}$S signal. The emission has a larger depleting effect, since the emission signal is 10.5 ‰, while the atmospheric signal is around 14 ‰, following Equation (9). Hence, emission introduces relatively light COS in the atmosphere, causing it to be depleted by –0.95 ‰ yr$^{-1}$. However, due to varying emission signatures, contrasting emission signals are expected over the oceans versus regions influenced by anthro-

pogenic emissions (Hattori et al., 2020; Davidson et al., 2021; Baartman et al., 2021). We carried out a sensitivity simulation with different emission signals that we discuss in Section 3.5.

Chemical production has a slightly depleting contribution (0.02 ‰ yr$^{-1}$), which depends on the signature of the two COS precursors and the fractionation during oxidation. $CS_2$, which is emitted at 10.4 ‰, leads to depleted atmospheric COS while





CH$_3$SCH$_3$, which is emitted at 20 ‰, enriches atmospheric COS.

Chemical loss has an enriching effect on COS, since the fractionation of all the oxidising reactions are negative. This means that the lighter isotope reacts away faster, leaving behind more enriched atmospheric COS ($\alpha_l < 1$ in Equation (9)). In the

troposphere this enrichment is about 0.11 ‰ yr$^{-1}$. In the stratosphere, chemical loss through photolysis leads to a strong enrichment (1.96 ‰ yr$^{-1}$). Figure 1 shows that the stratospheric COS indeed gets quite enriched. Note, however, that photolysis fractionation in the stratosphere is currently very uncertain. We therefore carry out a sensitivity test with different photolysis fractionations that are discussed in Section 3.5.

Transport to the stratosphere has an enriching effect on the tropospheric COS, though this effect is quite small (0.07 ‰ yr$^{-1}$). It has a larger, albeit a depleting effect on the stratospheric COS since lighter COS from the troposphere is mixed into heavier stratospheric air (1.96 ‰ yr$^{-1}$).

Lastly, dry deposition is a large sink for COS. The measured fractionation for this process is negative (Davidson et al., 2021).

Hence, biospheric uptake leads to a more enriched pool of atmospheric COS, enriching it by 0.79 ‰ yr$^{-1}$. Close to strong COS uptake regions, one would therefore expect to measure $^{34}$S-enriched COS.

### 3.5 Sensitivity Analysis

In this section, we study the uncertainties that are present in the COS isotopic signature and its associated fractionations, and how the subsequent sulfate $\delta^{34}S$ signature are affected by these uncertainties.

Figure 2 shows the sensitivity of the COS photolysis fractionation (J$_\epsilon$) and a case with no COS in the model. The outcome of a positive $\epsilon$ for the photolysis reaction of COS (J$_\epsilon$ = + 73 ‰ - blue, from Leung et al. (2002)) shows that the COS becomes strongly depleted in the stratosphere, compared to the base case (dashed black line). The heavier S moves faster to SO$_2$ and sulfate. Since COS oxidation in the stratosphere leads to a strong enrichment in the oxidation products SO$_2$ and sulfate, the sulfate at 18 km becomes too enriched (> 5 ‰) compared to Castleman et al. (1974)'s measurements. The 0 ‰ fractionation

(peach) case leads to sulfate that is slightly more enriched than the measured values, but remains still very close to the measurements. The negative $\epsilon$ (J$_\epsilon$ = − 10.5 ‰ - red from Lin et al. (2011)) shows that SO$_2$ and sulfate are enriched, but remain very similar to the base case. In conclusion, a small, negative $\epsilon$ is required to reproduce the SSA isotopic measurement of Castleman et al. (1974). A similar conclusion was drawn by Hattori et al. (2011) and Schmidt et al. (2012).

The case with no COS in the model (No COS - lighter blue) clearly shows that without COS as precursor, SO$_2$ and sulfate in the stratosphere would be strongly depleted in $\delta^{34}S$ and would not match the measurements reported by Castleman et al. (1974). This illustrates that the COS abundance and the COS photolysis fractionation control the $^{34}$S isotopic signature of SSA. Figure S10 shows the mixing ratios of COS, SO$_2$ and sulfate in a no COS case versus with COS present. While a small peak in sulfate in at 15 km is observed when there is no COS present, there is significantly less SSA formed compared to when





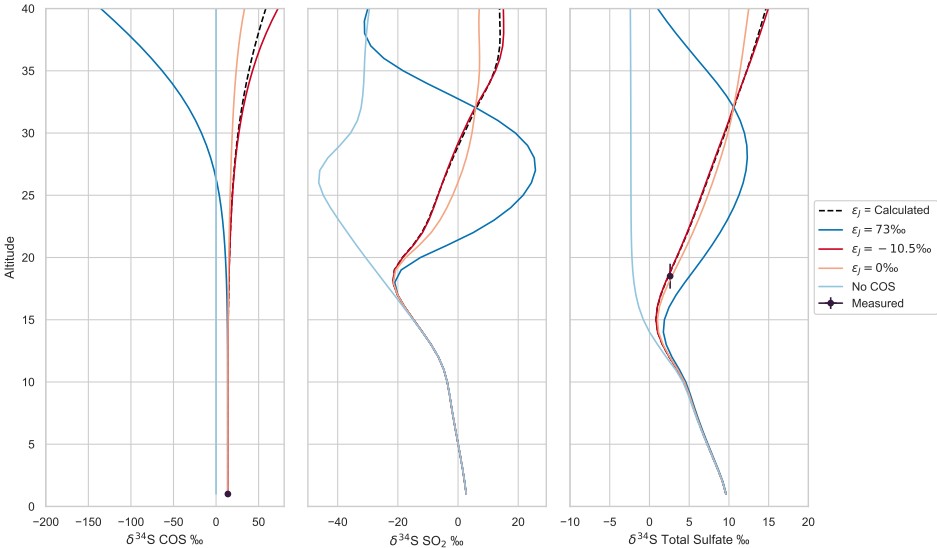

**Figure 2.** The $\delta^{34}$S [‰] propagation in height [km] for COS, SO$_2$ and sulfate. The dashed black line represents the base case presented in Figure 1. The ligher blue line (No COS) represents a case with no COS in the model. The blue line shows the sensitivity of the model to positive fractionation = 73 ‰ value for COS photolysis (Leung et al., 2002), the red line is for negative fractionation = −10.5 ‰ (Lin et al., 2011), and the peach line is for 0 fractionation.

COS is present. Stratospheric SO$_2$ is also less when there is no COS in the model.

The next sensitivity scrutinises the oxidation of CS$_2$ to COS. As CS$_2$ oxidation is a large source of COS (about 64% of total COS produced, see Table 7), a fractionation during this reaction would have significant effect on the COS isotopic signature.

In Section 4, we will discuss the considerations of isotopic fractionation in this oxidation pathway. Figure 3 shows the impact of adjusting the yield on the subsequent $\delta^{34}$S signature for COS, SO$_2$ and sulfate. When the yield towards CO$^{34}$S is lowered (pink), more of the lighter S ends up in the COS product. This is because less of the heavy S is ending up in COS and more in SO$_2$, while the yield for the $^{32}$S remains the same. Only in the stratosphere this signal propagates to SO$_2$ and sulfate, which get also slightly less enriched due to the lighter COS precursor. In contrast, when the yield is increased (purple line), the heavier S
propagates towards COS, and to stratospheric SO$_2$ and sulfate.

The top panel of Figure 4 shows the sensitivity of the subsequent $\delta^{34}$S signature for COS, SO$_2$ and sulfate to biosphere uptake fractionation ($\epsilon_{BIO}$ - green). For the biosphere, we show a fractionation range between 0 and −5 ‰, compared to −1.9 ‰ in our base scenario. As expected, no fractionation leads to lower $\delta^{34}S$ values and a fractionation of −5 ‰ leads to higher
$\delta^{34}S$ values. This range of fractionations encapsulates the sulfate measurements well. The effect on COS is small but there is a much more significant effect on SO$_2$ and sulfate in the stratosphere, depending on whether COS is more or less enriched.





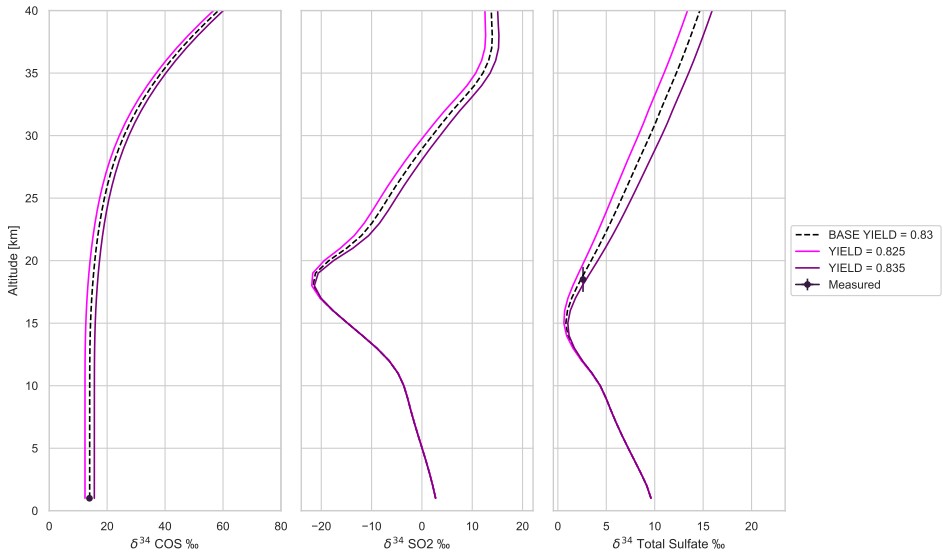

**Figure 3.** The $\delta^{34}$S [‰] propagation in height [km] for COS, SO$_2$ and sulfate. The dashed black line represents the base case presented in Figure 1. The pink line represents a case when the yield of the reaction CS$_2$ + OH is 0.825 towards COS and the purple shows a case where the yield is higher i.e. 0.835.

The bottom panel of Figure 4 shows the sensitivity to the emission signature of COS at emission (E$_{COS}$ - blue). Not surprisingly, results show that the more enriched the COS signature, the more enriched the SO$_2$ and sulfate will be in the stratosphere. Since the range of the emission signatures (from 8–14 ‰) is rather small, the effect is only a shift of about 1 ‰ for SO$_2$ and sulfate.

Again, for COS this signal appears throughout the atmosphere, while for SO$_2$ and sulfate, the effect is only observed in the stratosphere, where COS photolysis is the main source of SSA. Similarly to the biosphere case, the effect on COS is clear but there is a larger effect on SO$_2$ and sulfate in the stratosphere, because the COS is either more or less enriched.

Overall, this set of sensitivity simulations very clearly shows that COS is a source of stratospheric SO$_2$ and sulfate. The COS isotopic signature in the troposphere is important for the isotopic signatures of stratospheric SO$_2$ and sulfate. In the

troposphere, while the COS $\delta^{34}$S is different in each sensitivity test, the SO$_2$ and sulfate $\delta^{34}$S only respond in the stratosphere. This is especially the case when COS photolysis becomes significant, particularly above 16 km. The dominating factor that determines the $\delta^{34}$S signature of SSA is, therefore, the fractionation associated with COS photolysis.

## 4   Discussion

Severe gaps in knowledge exist concerning the sulfur isotopic budget in the atmosphere. In this study, we explored the sulfur

budgets and $\delta^{34}$S budgets for COS, SO$_2$ and sulfate in an attempt to integrate current knowledge of the system and performed





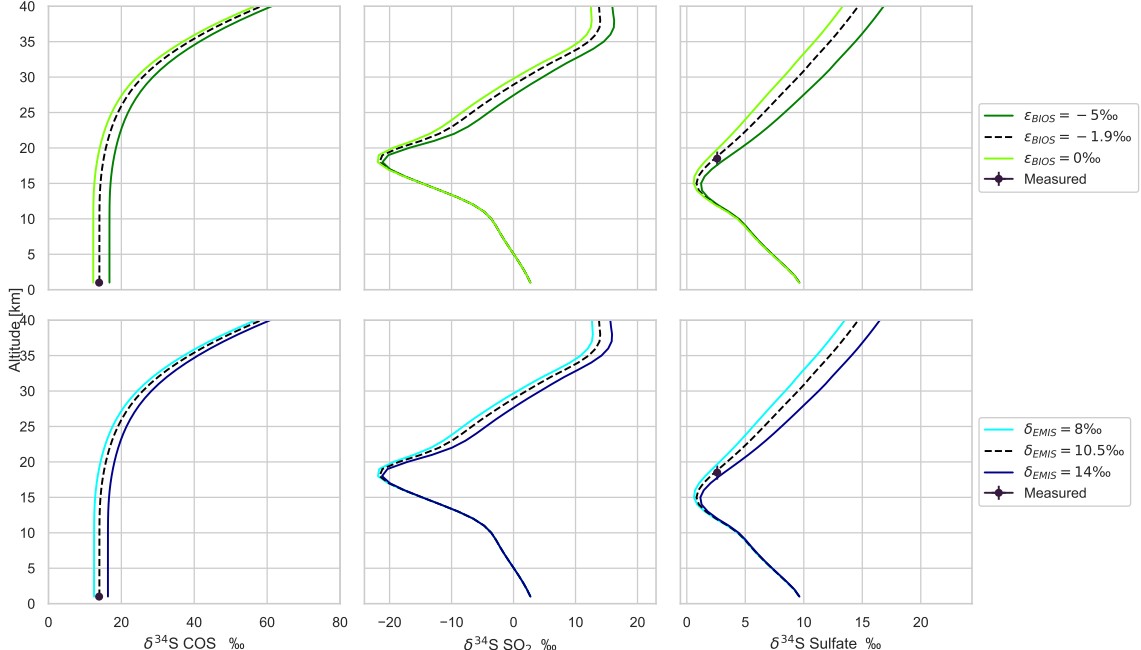

**Figure 4.** The $\delta^{34}S$ [‰] profile [km] for COS, $SO_2$ and sulfate. The dashed black line represents the base case presented in Figure 1. The green lines in the upper panel show the range of biosphere fractionations, with neon green showing a case of no fractionation and dark green showing a case of -5 ‰, as calculated by (Angert et al., 2019). The blue lines in the lower panel show the sensitivity of the model to different emission signatures, with cyan at 8 ‰, having a larger contribution from anthropogenic sources, and the darker blue at 14 ‰, more in line with marine measurements (Hattori et al., 2020; Davidson et al., 2021).

some sensitivity analyses to explore the uncertainties.

Firstly, it is important to realise that we use a horizontally-averaged column model. With such a modelling instrument, it is not possible to capture regional variability. A full 3D simulation of an isotope-enabled model is needed to indicate what hor-

5 izontal gradients in COS isotopologues are to be expected, specifically in the troposphere. For instance, above a forest, which acts as a sink for COS, we expect that measurements will be enriched in $\delta^{34}S$. It should be realized that the very small trend of +0.79 ‰ yr$^{-1}$ derived for dry deposition of COS (Table 9) represents a global average, and larger deviations are expected in the real atmosphere, in particular close to the surface.

10 A 1D column model also misses the atmospheric Brewer-Dobson circulation with upwelling motions in the tropics and downwelling in the higher/polar latitudes (Sheng et al., 2015). In our model we have a one-way net flux that always works counter gradient, which misses these effects due to the regional dynamics of the atmosphere. When modelling a two-way flux, it is estimated that about 90% of COS flows back to the troposphere (Sheng et al., 2015; Kremser et al., 2016). Yet, we are able





to adjust the transport such that a net COS loss in the stratosphere is calculated that is comparable to Sheng et al. (2015). Moreover, this study, for the first time, propagates the S-isotopic composition from S-emissions to SSA, and couples this to a (isotopic) mass-balance approach.

Secondly, much of the remaining uncertainties stem from the fact that the COS budget and its isotopic signature are not well constrained. The budget of COS is currently not closed, with missing sources that are still being studied. There are many hypotheses. Literature ranges from missing oceanic or anthropogenic sources, an overestimate of biosphere uptake, or missing chemical pathways (Whelan et al., 2018; Ma et al., 2021). While we do include the $CH_3SCH_3$ pathway in our model, which is often not considered in other models, we also include a missing source in our emission to reach the required mixing ratio

of COS in the troposphere. Figure 4 shows that more anthropogenic (marine) emissions lead to more depleted (enriched) COS $\delta^{34}S$.

The $\delta^{34}S$ isotopic signature of COS in the atmosphere is being investigated and recent studies expect the atmospheric signature to be between 9.7–14.5 ‰ (Kamezaki et al., 2019; Angert et al., 2019; Hattori et al., 2020; Davidson et al., 2021).

Moreover, current knowledge about fractionation to COS is mixed, ranging from the almost no information about COS production from $CS_2$ (discussed in more detail below), to the plethora of different $\epsilon$ values that have been calculated and measured for stratospheric COS photolysis. Understanding the photolysis fractionation would also help in constraining the COS isotopic signature in the stratosphere. The sensitivity test (Figure 2) highlights that a large, positive photolysis fractionation as posited by Leung et al. (2002) is not compatible with the SSA $\delta^{34}S$ measurements.

As discussed above, $CS_2$ oxidation is a major COS precursor that produces 280 Gg S yr$^{-1}$ of COS. Stickel et al. (1993) postulated a yield of 0.83 towards COS through oxidation, while the rest ends up as $SO_2$. As $CS_2$ has 2 S atoms, it is initially unclear what fraction of the $^{34}S$ atoms would propagate to COS. We can hypothesize what might happen to $^{34}SCS$ (we do not consider clumped isotopes in the model as these have very low abundances). The $CS_2 + OH$ reaction likely proceeds with the

formation of the $CS_2OH$-adduct, which propagates through $O_2$ addition to the central C-atom (Zeng et al., 2017). Following the proposed reaction path calculated by Zeng et al. (2017), isotope fractionation may happen either during the formation of the SCS(OH)-adduct or when $O_2$ attacks this adduct and progresses to COS and HOSO by C–S bond cleavage. OH might preferentially attach to the heavier S atom ($SC^{34}SOH$). In this case, the $O_2$ addition would result in increased formation of $^{32}SCO$ and $HO-^{34}S-O$ i.e. the $^{34}S$ would preferentially propagate to HOSO, leading to isotopically lighter COS. However,

thermodynamically, the C bond with the heavier S isotope should be more stable, hence cleavage is more likely when OH bonds to $^{32}S$, which would result in the $^{34}S$ atom ending up in COS. Hence, depending on which step is rate-determining, the produced COS could either be enriched or depleted compared to its $CS_2$ precursor. According to Zeng et al. (2017), the formation of the $SCS-OH$ adduct is rate-determining. This scenario represents a yield that is lower towards COS (0.825 in the sensitivity analysis) and hence would result in depleted COS $\delta^{34}S$, as seen in Figure 3.




The $CH_3SCH_3$ reaction pathway is also included in this study as a COS source. Despite the low yield, it still amounts to a significant source of COS due to the large amount of $CH_3SCH_3$ emitted. Whether this pathway does elicit some COS above pristine conditions is still highly uncertain, and it remains to be seen how this might affect the isotopic signature of COS. A recent study measured enriched $CH_3SCH_3$ emissions over the oceans, with an expectation that $CH_3SCH_3$ led to COS formation

(Davidson et al., 2021). We utilise this enriched $CH_3SCH_3$ (20 ‰) in our model, and it leads to an enriched COS pool in the atmosphere. Hence, over oceans not only do we expect emission of enriched COS, but also enriched COS from the $CH_3SCH_3$ pathway. Modelling studies have postulated $CH_3SCH_3$ oxidation as a missing source for COS (Lennartz et al., 2017; Ma et al., 2021).

Thirdly, the SSA [34]S signature that we use as a constraint for our model, is based on a study performed in the 1970s (Castleman et al., 1974). Our modelled sulfate profiles in the stratosphere show large gradients. There is therefore an obvious need for more measurements. Measurements of atmospheric COS isotopes, fractionation factors for COS photolysis, and the sulfate signal in the stratosphere and in volcanically quiescent periods would all provide valuable information.

Finally, in this paper we addressed only a steady state solution. The model was run for 60 years in order to achieve a steady state approximation for the S-isotopologues. The model can also be used to study transient, sporadic phenomena like volcanic eruptions. In a future study we intend to add emissions from a volcanic eruption to the stratosphere and then model the time-dependent removal of the sulfur species. Large, Plinian eruptions may add substantial amounts of sulfur to the stratosphere. This sulfur is then moved around the globe and after a few years is deposited at the Earth surface. Sulfate measurements in

polar ice show an anomalous, mass independent S-signature, which is considered to be a result of radiation effects due to a thick $SO_2$ plume (Baroni et al., 2007; Savarino et al., 2003; Gautier et al., 2018, 2019). These anomalous signals in [33]S and [36]S can be also modeled with PATMO, as we also include these isotopologues in our chemistry scheme. The needed sulfur chemistry framework is already present, as are the multi-frequency radiation calculations that are needed to resolve the $SO_2$ self-shielding that is expected in a volcanic plume (Ono et al., 2013; Lyons, 2007; Endo et al., 2015). This will be explored in

future work, in which we intend to study the effects of self-shielding on S-isotopologues of $SO_2$ in volcanic plumes (Lyons, 2007; Ono et al., 2013; Hattori et al., 2013).

## 5   Conclusion

Using a 1D column model we analysed the sulfur isotopic budget for a non-volcanic, modern day atmosphere. We modelled the isotopes of COS, $SO_2$ and sulfate in the atmosphere. To analyse the contributions to the isotopic composition of sulfur

species in steady state, we derived their $\delta$ budget equations and analysed the main processes that contribute to the modeled isotopic composition. We clearly demonstrate that COS is an important precursor for SSA during non-volcanic periods. We calculate that 77% of sulfur in SSA comes directly from COS. Concerning the [34]S isotopic signals, oxidation and photolysis in the stratosphere lead to a pool of enriched COS, which is transferred to $SO_2$ and SSA. In the troposphere, oxidation of $SO_2$





leads to isotopically depleted $SO_2$, but in the stratosphere $SO_2$ gets more enriched due to $SO_2$ production from COS photolysis. Hence, there is a large gradient in the $SO_2$ $\delta^{34}S$ profile in the lower stratosphere. In the stratosphere, the enriched isotopic signal is carried from COS to $SO_2$ and finally to sulfate. The modeled sulfate in the lowest part of the stratosphere matches well with the observations from Castleman et al. (1974) ($\delta^{34}S$ = 2.6 ‰) in volcanically quiescent times, at 18 km. The sulfur budget, and especially the isotopic budget of sulfur is, however, still not very well understood, with significant uncertainties present in the literature. In order to bridge this gap we also carried out a number of sensitivity analyses, which show that the isotopic composition of COS and fractionation processes of COS photolysis in the stratosphere are key factors that determine the $^{34}S$ isotopic composition of SSA in the lower stratosphere. Generally, positive (negative) $\epsilon$ values of COS photolysis result in less (more) enriched COS signals in the stratosphere. A negative, smaller $\epsilon$ brings the sulfate $\delta^{34}S$ in closer agreement with the only observation that is available. Therefore, there is definitely a need of additional measurements. Observations of COS isotopic signatures, COS photolysis factors, and sulfate isotopic signatures in the stratosphere would all lead to a better understanding of the stratospheric sulfur budget.

## 6 Author contribution

JN and MK were active in the conceptualization of the study and designed the methodology. JN performed the model simulations, carried out data analysis, and wrote the paper. MK, NNB, and TR supervised JN. SOD and MS developed the initial PATMO code and JN and MK carried out further code development to suit this project. JN, NNB, SOD, TR and MK contributed to editing the paper.

## 7 Acknowledgements

We acknowledge support from the Dutch Research Council (NWO) project 829.09.008. Maarten Krol was supported by the European Research Council (ERC) under the European Union's Horizon 2020 research and innovation programme under grant agreement No 742798.



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
