# Peer review of "Modelling the atmospheric 34S-sulfur budget in a column model under volcanically quiescent conditions"

_Atmospheric Chemistry and Physics, 2022_

## Referee Comment (RC1)

**Review of "Modelling the atmospheric $^{34}$S-sulfur budget in a column model under volcanically quiescent conditions" by Juhi Nagori et al.**

The study attempts to include $^{34}$S/$^{32}$S isotope fractionation of sulfur species and of the chemical, physical and biological processes affecting them in an atmospheric sulfur model. It does so by a simple 1D column model approach and concentrates on key sulfur species of atmospheric relevance.

In general, this is an interesting concept and a useful approach that should be developed further by all means. However, at this point and with the model and information used, the study falls short of addressing the questions given in the introduction and, overall, to provide new information that goes beyond the current state of the art. In my opinion, results and conclusions are to a large extent imported as *a priori* information that goes into the model or by nudging parameters in the model to make key numbers agree with previous literature. I try to explain what I mean for each of the three research questions given at the end of the introduction:

*1. What is the COS contribution to SSA?*

First of all, I don't think there is still much debate about the contribution of carbonyl sulfide to SSA in volcanically quiescent times (see my comment regarding page 2, lines 16 – 23 further below). But even if there was, I don't see that this study provides any new knowledge or arguments in that respect.

In Section 2.1 (paragraph starting on page 5, line 17) it is stated that the eddy diffusion coefficient is doubled in order to match the stratospheric COS turnover of 40 Gg S yr$^{-1}$ reported by Brühl et al. (2012). If I understand this correctly, you nudge your model to simulate exactly the stratospheric COS turnover given by Brühl et al. (2012), right? By doing so, you essentially prescribe the COS flux to the stratosphere. And because the stratospheric chemistry included in your model will inevitably turn any COS reaching the stratosphere into SO$_2$ and subsequently SSA, the absolute COS contribution to SSA is fixed at 40 Gg S yr$^{-1}$. The relative contribution of 77 % to SSA depends on how much tropospheric SO$_2$ your model transports to the stratosphere, and for that, you can hardly expect to obtain better or more realistic numbers than the two studies by Brühl et al., 2012, and Sheng et al., 2016, that use full atmospheric models.

Therefore, other than your statement "*In the next section we explore the full stratospheric sulfur budget, to investigate the importance of COS for SSA formation.*" (page 16, line 25) suggests, there is little "exploration", and the result with respect to the COS contribution to SSA given in Section 3.3 (Page 17, lines 1 – 4) is, to a large extent, prescribed. Therefore, when you write "*We clearly demonstrate that COS is an important precursor for SSA during non-volcanic periods*" (page 23, line 31, in your conclusions), I consider this hardly a conclusion but rather a feature of your model that is directly imported from previous knowledge.

*2. How can isotopic information help constrain the sulfur budget?*

Here, I'm not sure what exactly you mean by "*the sulfur budget*". By focusing mainly on the sulfur compounds contributing to stratospheric sulfur (i.e. SO$_2$, H$_2$SO$_4$ and COS), and in the troposphere only really discuss a *budget* when it comes to COS, you are obviously not looking at the global sulfur cycle at large.

As explained above, the stratospheric sulfur budget appears to be fixed in your model. You do, for the first time, simulate a corresponding sulfur isotope budget, but with available knowledge and observations, that does not tell us much, except that a large positive fractionation for COS photolysis proposed by some studies is probably unrealistic (cf. some of my specific comments below).

With respect to the COS budget, a number of studies have recently addressed this question (Angert et al., 2019; Hattori et al., 2020; Davidson et al., 2021), so the idea is hardly new. And while your approach bears some potential to constrain the COS budget and processing in the stratosphere, the 1D approach is not suitable to go beyond the mass balance approach employed in these studies that also consider regional and temporal variabilities to isolate the fractionation for individual COS sources and sinks (also see my specific comment below regarding page 19, line 3 to page 20, line 12).

I'm not saying that the use of isotopic information isn't valuable, and at the end of your conclusions, you provide a useful discussion of this question. But this is more an outlook rather than anything conclusive: large uncertainties in the fractionation parameters of the many processes involved and too few observations, particularly in the stratosphere, prevent your 1D model approach from really adding new knowledge or reducing uncertainties. With more information on the fractionation of sources, sinks and processes, and more observations of sulfur isotope ratios in the atmosphere, there is certainly a lot of potential (although I'm not convinced that the 1D approach is the most efficient one, and that it is not worth the – probably substantial – extra effort implementing isotope fractionation in a full 3D atmospheric model).

*3. What are the largest uncertainties in the COS isotopic budget?*

Uncertainties in the COS isotopic budget are discussed in your introduction, methods, and in the context of your sensitivity studies. From what I can see, the discussion adequately reflects the state of the art (i.e. the information available from the literature) and in your sensitivity runs, you encompass most of these uncertainties.

Because they are all taken from the literature, the uncertainties themselves are *a priori* rather than *a posteriori* knowledge in your study. And because the combined uncertainties related to different terms in your model offer more degrees of freedom than you have constraints (e.g. from observations), you can hardly draw any conclusions with respect to the significance or importance of individual uncertainties (again, see my specific comment below regarding page 19, line 3 to page 20, line 12).

Compared to the previous studies by Brühl et al. (2012) and Sheng et al. (2016), I don't see that your 1D column model approach offers any new information in terms of science, and I would hardly call it "*pioneering*" (page 4, line 2) in spite of the fact that the isotope information is added as a novel feature. Therefore, I do not consider the manuscript publishable in ACP in its present form.

This does not mean that the work that you put into implementing and parameterizing the isotope information in a model isn't useful and important, and I strongly encourage you to rewrite and publish it as a proof-of-concept-study, for example as an ACP Technical Note or as a GMD Article. The results on the COS photolysis fractionation (where you can reduce uncertainties to some extent, see some of my specific comments below) could still be included to show the potential when the necessary information (in this case the SSA fractionation observed by Castleman et al., 1974) is available.

Below, I include a number of specific comment that either illustrate my criticism above further, and/or may help you in rewriting the manuscript.

**Specific comments:**

Page 1, line 17: It should be "stratospheric sulfate aerosol" (you should consistently use the same term throughout the paper)

Page 2, line 2: This statement is somewhat inaccurate. COS is the most abundant reduced sulfur compound in the troposphere (at all times) and it is the largest contributor to SSA in volcanic quiescent times.

Page 2, line 5: The expression "above the ozone layer" is not accurate. COS photolysis can occur only when there is sufficient UV radiation, and if you look at the stratospheric COS distribution (e.g. in Barkley et al., 2008) and at the modelling study by Brühl et al. (2012), then it is most relevant in the tropics above ~ 20 km altitude.

Page 2, lines 7 – 8: I suggest to elaborate more on the open questions regarding COS sources and sinks; "*still poorly constrained*" is not a precise term, and personally, I would not use it in the context of the COS budget. Issues related to the COS budget were also discussed in some detail by Kremser et al. (2016), and I think we have come one or two steps further (towards better constraining the budget) since that review was published.

Page 2, lines 13 – 14: Again, this statement is inaccurate. In current models, a 0.7 % yield is assumed everywhere. The older laboratory studies that you cite already hypothesized that this yield could be zero in polluted conditions (based on efficient quenching by NOx), and some rather recent evidence (e.g. Jernigan et al., 2022) shows that COS production from DMS could potentially (!) be even higher than 0.7 % under "extremely pristine conditions".

Page 2, lines 16 – 23: For the first sentence, Notholt et al. (2003) is not a good reference, because the statement was merely an introductory sentence in that paper. The first to suggest this was, in fact, Crutzen (1976). And I think we can now say that COS is considered not only a major but the most important source of SSA under volcanically quiescent conditions (Kremser et al., 2016, and references therein, which are the ones you also cite later in the paragraph). I really don't think there is still much debate about the contribution of COS to SSA in volcanically quiescent times (that concerns also the same statement in your abstract on page 1, line 4). While the Kjellström (1997) reference was important at the time (25 years ago) by showing that the net COS flux to the stratosphere was probably lower than some people thought and that some $SO_2$ could at least make it to the lower stratosphere, there now exist comprehensive studies (cited in your manuscript) that give us a pretty good overall picture, even though there are still some uncertainties on the exact numbers.

Page 3, line 6: Given the evidence in the work cited (in particular Hattori et al., 2020), "*it is expected that anthropogenic COS is likely to be more depleted than oceanic COS*" is probably an understatement. I suggest to make this statement more affirmative. And it would be logical to discuss the role of COS sink processes already here: what do we know about the fractionation related to the vegetation and soil sinks on land? As a side note in this context, I'm not convinced that I we can expect a clear ocean source signature with one single number for $\delta^{34}S$. For direct COS exchange between the ocean and the atmosphere, while the ocean is a net source, regional, diurnal and seasonal fluxes vary and are often enough negative because of a complex interplay between COS production and removal processes in the water. The magnitude of the net flux varies with region and season, and I would expect the net isotope fractionation effect to so as well.

Page 3, lines 11 – 12: Better write "*...within the range of -5 to 0 ‰ calculated by Schmidt et al. (2012) for this reaction.*"

Page 3, lines 14 – 19: This is a good discussion of the large uncertainty range of COS photolysis. This uncertainty (here, the term "*not well constrained*" would be more than appropriate) of the key parameter for the fractionation of SSA produced from COS is absolutely worthy to be addressed, and I see it as one of the few questions where your modelling study in its present form can make a significant contribution to reduce uncertainties (see comments further below and my general comments above).

Page 4, lines 10 – 11: This sentence is odd and "*explore*" does not seem to be the right word here. After all, you designed the model and parameterized the sulfur chemistry.

Page 4, lines 15 – 16: these uncertainties are indeed rather large. In my opinion, they give the model far too many degrees of freedom for it to produce meaningful outcome, and you do not have enough information to resolve uncertainties except for the fractionation of COS photolysis.

Page 4, lines 21 – 23: How do you treat the vast spatial and temporal differences? Do you just average everything into one global number?

Page 5, lines 4 – 6: When I read this, my first feeling is that this approach oversimplifies things and may not fully represent the real processes in the atmosphere, e.g. in terms of contribution from different oxidation pathways. It is clear that with the 1D approach, you do not account for regional differences, but does your model account for seasonal and diurnal variability?

Page 5, line 19 – page 6, line 3: I suspect that the tuning of the eddy diffusivity to nudge the model to the stratospheric COS turnover given in the previous literature (cf. my general comment above) is needed to account for tropical upwelling that obviously is important for the upward COS flux in the full 3D models. If I understand this correctly, then this means that, in terms of vertical transport, you are tuning your 1D model to represent a tropical regime, while in terms of photochemistry, it simulates more or less mid-latitude conditions, i.e. with the standard US oxidant profiles (page 4, line 5) and using a mean zenith angle of 57.3 ° (page 6, line 25).

Page 6, lines 25 – 27: Continuing from my last point above, I don't find the overall setup convincing, because the overall rate balance between photolytic processing and ascend/mixing in the 1D model does not, as far as I can see, represent anything I would expect in the real atmosphere, which in turn poses questions with respect to the results, at least when you interpret them quantitatively.

Page 9, lines 6 – 7: Please give the references that your assumption for these numbers is based upon!

Page 9, lines 12 – 13: Is the overhead burden of sulfur gases really a key factor here? I would expect overhead ozone to be the more important factor determining photolysis rates.

Section 2.2: If I understand this correctly, you combine all COS sources and sinks into one integrated global source and one sink (that you term "*dry deposition*"), each with one respective average $\delta^{34}S$ that you derive from literature information on isotope signatures and source split. With the purpose of trying to keep things simple, this approach is fine, but it makes me wonder how you can then possibly derive and information or constraints on the COS budget with your model? My impression is that any information on the COS budget and isotope effects related to the budget is taken from the recent literature.

Section 3.1, first paragraph, top panel of Fig. 1: While you investigate in some detail how well your model fits observed isotope ratios, there is no comparison for the mixing ratio profiles. This is surprising, because representative profiles can be derived from a wealth of satellite data. With such a comparison, you could test if your model really does represent global averages, and how it compares to real profiles when different latitude bands and seasons are considered. That could set the scene for looking at observed isotope ratios, which I consider meaningful only if the concentrations are also reasonably well represented.

In other words: why would you trust the simulated isotope fractionations and compare them to observations if you are not confident that the mixing ratios are represented well?

Page 15, lines 8 – 9: Here, it would be interesting to state how much SSA at 18.5 km originates from COS and the tropospheric $SO_2$ source in your model (when looking at the top panels of Fig. 1, both seem to be significant, and based on your Figure S10, the tropospheric $SO_2$ contribution dominates below 20 km). If there is still a significant contribution from $SO_2$, then this would add yet another uncertain tuning parameter (that is independent on COS) when it comes to bring your model close to the observed $\delta^{34}S$.

Page 16, lines 3 – 5: This statement is even more inaccurate, unclear and confusing than the earlier one (page 2). If you use the 0.7 % as one global number, then simply say so, cite the references and leave it at that!

Page 16, line 8: What do you mean by "*adjusted*"? First you include an "*additional source*" that was introduced to compensate for the larger flux to the biosphere, and then you adjust the flux to the biosphere in your model. To me, that does not make sense. And why did you adjust to 527 ppt and not 500 ppt?

Page 16, lines 9 – 11: Could this be an effect of the OH profile chosen in your 1D model, and that in other models and the real atmosphere, OH and other compounds as well as physical conditions vary in a complex manner?

Page 16, line 19: Note that this is a little less than half the burden of 283.1 Gg S given in Sheng et al. (2015), which is in excellent agreement with a burden of 280 Gg S derived from satellite observations (Kloss, 2017).

Page 18, lines 6 – 8: This is true, and the sensitivity test for the photolysis reaction is the one point where I see this study making a contribution towards reducing uncertainties.

Page 18, lines 10 – 12: The described transport enrichment is a response to $\delta^{34}S$ gradients in the model at steady state and the model does not include any *mechanistic* fractionation during transport, right?

Page 18, line 19: This should be "*...the subsequent sulfate $\delta^{34}S$ signature is affected...*"

Page 18, lines 20 – 28: As stated above, the sensitivity study on the photolysis fractionation appears to yield a viable result, although it is to some extent compromised by the likely sensitivity of the Castleman et al. (1974) observations towards tropospheric $SO_2$ (at least in your 1D model, see my comment above related to page 15, lines 8 – 9).

Page 19, line 3 to page 20, line 12: These sensitivity tests yield expected, almost trivial, results. I wonder, what we can really learn from them. What you show is that the fractionation of stratospheric $SO_2$ and sulfate respond to the fractionation of tropospheric COS, which is expected and rather trivial. And when you state that the "*set of sensitivity simulations very clearly shows that COS is a source of stratospheric $SO_2$ and sulfate*" then, again, this is a direct result of the design of the model with a prescribed COS flux to the stratosphere. Clearly, this does not in any way go beyond what was already known. And when it comes to looking at uncertainties in the various fluxes, reaction rates and fractionation effects, you have at least three rather uncertain parameters, and within the margins explored in your sensitivity runs, they all have very similar effects (e.g., the top and bottom panels in Figure 4 look almost exactly the same, except for the colors and legends), so the comparison to stratospheric observations does not help to reduce any of these uncertainties. This would be true even if you had perfectly accurate profiles of observed $\delta^{34}S$ in stratospheric $SO_2$ and sulfate. Clearly, to explore and hopefully reduce uncertainties of

the various parameters affecting $\delta^{34}S$ of COS in the atmosphere, you need to look at regional and temporal variabilities in order to separate the effects of individual sources and processes, and a 1D model is not the right tool to do so.

Page 20, line 15: I'd say the current knowledge of the system was imported rather than integrated (see my general comments)!

Page 22, lines 18 – 19: In my opinion, that is the one viable conclusion that can currently be drawn from your study. And even that depends on the accuracy of an average $\delta^{34}S$ in SSA from a few measurements in the 1970s between 18 and 19 km, where, at least in your model, the contribution from tropospheric $SO_2$ is large if not dominant (cf. my other comments above).

Page 22, lines 21 – 34: This is an interesting discussion on its own, but it does not help the overall purpose of the paper. And besides the fractionation of the $CS_2 \rightarrow COS$ reaction, the $\delta^{34}S$ of the $CS_2$ source may be equally or even more important.

Page 23, line 7: At least for Lennartz et al. (2017), it is not true that DMS oxidation was "*postulated*" as the missing COS source. It was rather not ruled out as one possibility to explain part of the missing source, but more arguments were given against than for this possibility.

**References not already listed in the Nagori et al manuscript:**

Barkley, M. P., Palmer, P. I., Boone, C. D., Bernath, P. F., and Suntharalingam, P.: Global distributions of carbonyl sulfide in the upper troposphere and stratosphere, Geophys. Res. Lett., 35, L14810, https://doi.org/10.1029/2008GL034270, 2008.

Jernigan, C. M., Fite, C. H., Vereecken, L., Berkelhammer, M. B., Rollins, A. W., Rickly, P. S., et al.: Efficient production of carbonyl sulfide in the low-NOx oxidation of dimethyl sulfide. Geophys. Res. Lett., 49, e2021GL096838, https://doi.org/10.1029/2021GL096838, 2022.

Kloss, C.: Carbonyl Sulfide in the stratosphere: airborne instrument development and satellite based data analysis, Ph.D., Chemistry Department, Bergische Unviersität Wuppertal, Wuppertal, http://elpub.bib.uni-wuppertal.de/servlets/DocumentServlet?id=7570&lang=en, 2017.

---

## Author Comment (AC1)

We thank the reviewer for their valuable feedback on our paper. We respond here to some of the larger comments the reviewer has detailed in their review.

The reviewer makes some valid and valuable points in reference to our article. However, we feel that he misjudges the value of the paper. While we agree that some of the results of the paper are not ground-breaking and new, they do confirm the existing literature with a complete sulfur chemistry scheme, which we then extend to isotope modelling. It is a first attempt at fully incorporating the isotope scheme in a 1D model, with all the S-molecules included. This an important step before incorporating the isotope scheme in a 3D model, in which the spatial and regional effects can be better scrutinized. We would like to point out that modelling of S-isotopologues in atmospheric transport models is not a trivial task, and we think that this 1D modelling study is a valid first step.

Regarding the COS contribution to SSA; we are limited in the case of SSA with a lack of isotopic measurements, however this issue would also be a problem in a 3D model. We do agree that for $SO_2$ and COS there are enough measurements available (e.g. satellite data) that we can add in our paper to better validate it. We would like to add some Michelson Interferometer for Passive Atmospheric Sounding (MIPAS) satellite observations in the case of COS (Glatthor et al., 2017, Ma et al., 2021).

Furthermore, relating to the point of reviewer 1 on the budget of sulfur isotopes in the stratosphere, we would like to point that the highlight of our paper comes from the isotopic aspect of it, where we are able to show the vertical profiles of the sulfur gases in the stratosphere and troposphere and calculate the contribution to the isotopic budget. For instance, what we notice in the sulfate isotopic budget is that sedimentation makes the pool of stratospheric S enriched by removing the lighter S. Therefore, it is definitely an interesting finding to see that all the S compounds in the stratosphere are enriched in this model. The isotopic budget calculations are also not trivial, as they describe the contribution of each process to the isotopic signature of each molecule in the atmosphere, something that is worthwhile to implement in a 1D model first before moving to a full 3D implementation. We feel that these novel aspects of our paper are not valued enough. We are of course willing to restructure the paper so as to better highlight the isotopic aspect of the paper.

To conclude, we agree with some of the more substantive remarks of this reviewer. However, the reviewer clearly undervalues some of the novelty of the paper. We think that – with the restructuring we propose above – the paper provides enough interesting material for the scientific community.

References
Glatthor, N., Höpfner, M., Leyser, A., Stiller, G. P., von Clarmann, T., Grabowski, U., Kellmann, S., Linden, A., Sinnhuber, B.-M., Krysztofiak, G., and Walker, K. A.: Global carbonyl sulfide (OCS) measured by MIPAS/Envisat during 2002–2012, Atmos. Chem. Phys., 17, 2631–2652, https://doi.org/10.5194/acp-17-2631-2017, 2017. a, b, c, d, e, f

Ma, J., Kooijmans, L. M., Cho, A., Montzka, S. A., Glatthor, N., Worden, J. R., ... & Krol, M. C. (2021). Inverse modelling of carbonyl sulfide: implementation, evaluation and implications for the global budget. *Atmospheric Chemistry and Physics*, *21*(5), 3507-3529

---

## Author Comment (AC2)

We thank the reviewer for their valuable feedback on our paper. We respond here to some of the larger comments the reviewer has detailed in their review.

Our model is a first attempt at fully incorporating the isotope scheme in a 1D model, with all the S-molecules included. This an important step before incorporating the isotope scheme in a 3D model, in which the spatial and regional effects can be better scrutinized. We feel that this effort is not valued enough by this reviewer.

The spectral resolution used in the model is a key feature to model photochemically-induced isotopic effects. In this study the Schumann-Runge bands are not central except when strong contributions from stratospheric $SO_2$ photolysis by volcanic emissions are considered. The reviewer quite correctly points out that the spectral resolution is needed to calculate isotopic imprint during photolysis created by the red shift in the spectra. The spectral features of the solar flux are also accounted by the spectral resolution implemented in the model and we can explain this better in a future version of the manuscript.

The Rayleigh scattering is a feature that was not considered relevant and it could be added to the opacity term in eq. 4 (in the manuscript). However, there are two main features that render the scattering effect irrelevant at the stratospheric altitudes at which we evaluate photolysis.

First, the effect of Rayleigh scattering on the COS photolysis rate. The absorption cross section of COS at 224—225 nm (peak) is $3 \times 10^{-19}$ cm$^2$. At the same wavelength the Rayleigh scattering cross section is $1.515\text{-}1.259 \times 10^{-25}$ cm$^2$ (Bucholtz, 1995). For comparison, the cross section of $O_2$ at the same wavelength is $0.5 \times 10^{-23}$ cm$^2$. There is a 6 orders of magnitude difference between the Rayleigh and the COS spectrum and 2 orders of magnitude than the $O_2$ Herzberg continuum (which is accounted for in eq. 4 opacity term). Hence, we do not expect for the Rayleigh scattering to be of any significance to the stratospheric COS photolysis.

Second, any molecule absorbing or scattering with a cross-section several orders of magnitude smaller the absorption cross-section of the molecule in question, could potentially inflict a significant isotopic effect. This only happens if the spectrum (Rayleigh scattering in this case) has a pronounced spectral structure (just like the Schumann-Runge bands). However, the Rayleigh scattering spectrum only shows very small changes over the wavelengths that overlap with the COS absorption cross-sections.

To verify the neglect of Rayleigh scattering on the photolysis, we also looked at the actinic flux in the Tropospheric Ultraviolet and Visible (TUV) radiation model (NCAR, 2022) and especially the difference between the direct flux and diffuse radiation between 200—300 nm between altitudes of 10—40 km. These ranges were looked at since we are concerned about the stratospheric radiation and photolysis of COS, which is significant only above 15 km. Above 40 km no COS is left to consume (See Figure 1 in manuscript). The wavelength range contains the COS absorption peak (224—225 nm). Below 200 nm oxygen and ozone absorption dominate (Molina & Molina, 1986).

[Figure]

*Figure 1: Actinic flux calculated (left) between 10—40 km between 200—300 nm where COS photolysis takes place. (Right) Percentage of this actinic flux that is direct (solid lines) versus diffuse (dashed lines) radiation between 200—300 for 5 different heights shown in different colours.*

First, the actinic flux is much higher above 280 nm as most of the absorption takes place at shorter wavelengths. We can see that at 30 km (where COS is efficiently removed) the direct radiation dominates (~98%) the diffuse radiation. This is what led us to not include Rayleigh scattering in our model, as we are mostly concerned with photolysis of COS in the stratosphere as it is considered the major source of SSA formation.

The reviewer also mentions that "Replacing a diurnal average of photolysis rates by using a calculation with an average zenith angle screws up the altitude dependence", yet we are able to reproduce the COS mixing ratio with altitude, though we agree that can better show this by using COS satellite observations.

Thus, in order to further improve the manuscript, we also will add observations that exist for COS mole fractions. For COS profiles, there are satellite observations like the Atmospheric Chemistry Experiment Fourier Transform Spectrometer (ACE-FTS) and the Michelson Interferometer for Passive Atmospheric Sounding (MIPAS). Though the shape of the COS profile looks similar to satellite observations (see: Ma et al., 2021), we would include the MIPAS observations since we are concerned about reproducing the upwelling of COS to the stratosphere in the tropics in order to achieve the 40 Gg S yr[-1] stratospheric loss of COS (Glatthor et al., 2017, Ma et al., 2021). ACE-FTS, operating in solar occultation, has less sensitivity to the tropics.

To conclude, we agree with some of the more substantive remarks of this reviewer. However, the reviewer clearly undervalues some of the novelty of the paper and/or misjudges the

scrutiny we attempted in writing the manuscript. We think that – with the additions we propose above – the paper provides enough interesting material for the scientific community.

References
Bucholtz, A. (1995). Rayleigh-scattering calculations for the terrestrial atmosphere. Applied Optics, 34(15), 2765-2773.

Glatthor, N., Höpfner, M., Leyser, A., Stiller, G. P., von Clarmann, T., Grabowski, U., Kellmann, S., Linden, A., Sinnhuber, B.-M., Krysztofiak, G., and Walker, K. A.: Global carbonyl sulfide (OCS) measured by MIPAS/Envisat during 2002–2012, Atmos. Chem. Phys., 17, 2631–2652, https://doi.org/10.5194/acp-17-2631-2017, 2017. a, b, c, d, e, f

Ma, J., Kooijmans, L. M., Cho, A., Montzka, S. A., Glatthor, N., Worden, J. R., ... & Krol, M. C. (2021). Inverse modelling of carbonyl sulfide: implementation, evaluation and implications for the global budget. *Atmospheric Chemistry and Physics*, *21*(5), 3507-3529

Molina, L. T., & Molina, M. J. (1986). Absolute absorption cross sections of ozone in the 185-to 350-nm wavelength range. *Journal of Geophysical Research: Atmospheres*, *91*(D13), 14501-14508.

NCAR (Retrieved 2022), TROPOSPHERIC ULTRAVIOLET AND VISIBLE (TUV) RADIATION MODEL, Retrieved: https://www2.acom.ucar.edu/modeling/tropospheric-ultraviolet-and-visible-tuv-radiation-model

---

## Author Comment (AC3)

We received two reviews of our paper and have submitted our replies recently. As you can read in our reply, we feel that the value of our paper was misjudged.

Reviewer 1 appreciates the paper, but simply thinks the added value is limited, a viewpoint we do not share. It is the first scientific publication in which a complete and consistent sets of isotope fractionations in sulfur compounds has been implemented, and the novelty of the paper is that we combine the different S compounds and their isotopic signals using isotope effects in all relevant reactions, from the troposphere to the stratosphere. More details are mentioned in the reply to reviewer 1.

Reviewer 2 mentions that the photolysis calculation ignores the Rayleigh scattering which has a large impact on altitude and wavelength dependent photon fluxes. However, when we check the scattering cross-sections and compare the diffuse versus direct radiation at altitudes where COS photolysis is important, we see that Rayleigh scattering is of minor importance. Including Rayleigh scattering will thus not change our results.

We are very willing to add material to address the minor suggestions, like better highlighting the new findings, and a validation in the stratosphere with satellite data. We also aim to clarify details like the solar radiation description (why Rayleigh scattering is not included), and highlight the isotopic results better.
We think the paper constitutes a consistent piece of work that connects the S-isotope composition in the troposphere to SSA in the stratosphere, something that has not been presented before, and is therefore worth publication.

We therefore hope that the paper will be accepted, and we are happy to provide the additional modifications that we propose in our rebuttal,